# Exact Boolean Abstraction of Linear Equation Systems

**Emilie Allart** 1,2,†⬤, **Joachim Niehren** 1,3,†⬤ and **Cristian Versari** 1,2,*,†

1 CRIStAL—Centre de Recherche en Informatique, Signal et Automatique de Lille—UMR 9189, Université de Lille—Campus Scientifique, 59655 Villeneuve-d'Ascq, France; emilie.allart@univ-lille.fr (E.A.); joachim.niehren@inria.fr (J.N.)
2 Faculte des Sciences et Technologies, University of Lille, 59650 Villeneuve-d'Ascq, France
3 Inria, Université de Lille, 59000 Lille, France
* Correspondence: cristian.versari@univ-lille.fr
† These authors contributed equally to this work.

**Abstract:** We study the problem of how to compute the boolean abstraction of the solution set of a linear equation system over the positive reals. We call a linear equation system $\phi$ exact for the boolean abstraction if the abstract interpretation of $\phi$ over the structure of booleans is equal to the boolean abstraction of the solution set of $\phi$ over the positive reals. Abstract interpretation over the booleans is thus complete for the boolean abstraction when restricted to exact linear equation systems, while it is not complete more generally. We present a new rewriting algorithm that makes linear equation systems exact for the boolean abstraction while preserving the solutions over the positive reals. The rewriting algorithm is based on the elementary modes of the linear equation system. The computation of the elementary modes may require exponential time in the worst case, but is often feasible in practice with freely available tools. For exact linear equation systems, we can compute the boolean abstraction by finite domain constraint programming. This yields a solution of the initial problem that is often feasible in practice. Our exact rewriting algorithm has two further applications. Firstly, it can be used to compute the sign abstraction of linear equation systems over the reals, as needed for analyzing function programs with linear arithmetics. Secondly, it can be applied to compute the difference abstraction of a linear equation system as used in change prediction algorithms for flux networks in systems biology.

**Keywords:** linear equation systems; abstract interpretation; program analysis; systems biology

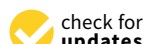



## 1. Introduction

We develop approaches to remedy the incompleteness of abstract interpretation [1] of linear equation systems over the reals, in the algebra of booleans $\mathbb{B} = \{0, 1\}$ and the structure of signs $\mathbb{S} = \{-1, 0, 1\}$. These abstractions have various applications: in systems biology, the boolean abstraction underlies abstractions of chemical reactions networks into Boolean networks [2,3]. In program analysis, the sign abstraction can be applied to functional programs with arithmetics for analyzing the signs of the possible values of floating-point variables [4,5].

The soundness of abstract interpretations of first-order logic formulas without negation was shown by John [6–9]. It applies to the interpretation in any concrete structure $S$, as long as it is connected by a homomorphism $h : S \to \Delta$ to the abstract structure $\Delta$. The concrete interpretation of a first-order formula $\phi$ is the set of concrete solutions $sol^S(\phi)$, and its abstract interpretation is the set of its abstract solutions $sol^\Delta(\phi)$. John's soundness theorem (see Theorem 1 below) states that the set of abstract solutions of overapproximates the abstraction by $h$ of the set of concrete solutions:

$$h \circ sol^S(\phi) \subseteq sol^\Delta(\phi)$$

When choosing the operators in $\Sigma_{bool} = \{+, *, 0, 1\}$, the class of negation-free first-order formulas with operators in $\Sigma_{bool}$ extends on the classes of linear and polynomial equation systems. In this article, we consider the boolean abstraction which is the unique homomorphism $h_{\mathbb{B}} : \mathbb{R}_+ \to \mathbb{B}$, and the sign abstraction which is the unique homomorphism $h_{\mathbb{S}} : \mathbb{R} \to \mathbb{S}$ with respect to the operators in $\Sigma_{bool}$. The boolean abstraction maps any strictly positive real to 1 and 0 to 0. The sign abstraction extends on the boolean abstraction while mapping all strictly negative reals to $-1$. We note that the structure of signs $\mathbb{S}$ is *not* an algebra since the sum of a positive and a negative number may have any sign.

### 1.1. Problematics

A natural question is whether abstract interpretation is complete [10] for the abstraction of formulas induced by a homomorphisms $h : S \to \Delta$, i.e, whether for all negation-free first-order formulas $\phi$ with the same operators:

$$h \circ sol^S(\phi) = sol^\Delta(\phi)$$

We call a formula $\phi$ $h$-exact if it satisfies this property. A counter example against the completeness of abstraction interpretation for the boolean and the sign abstraction is the linear equation $\phi_0$ equal to $x + y \stackrel{\circ}{=} x + z$. Here, we write $\stackrel{\circ}{=}$ for the equality symbol inside the logic, to point out its difference from equality in the language of mathematics. Formula $\phi_0$ is neither $h_{\mathbb{B}}$-exact nor $h_{\mathbb{S}}$-exact. This can be seen as follows. Over the reals $\phi_0$ is equivalent to $y \stackrel{\circ}{=} z$, so that all assignments $\tau$ that are abstractions of concrete solutions of $\phi_0$ must satisfy $\tau(y) = \tau(z)$. When interpreted abstractly over $\mathbb{B}$ or $\mathbb{S}$, however, $\phi_0$ admits the abstract solution $\tau = [x/1, y/1, z/0]$ which is not the abstraction of any concrete solution since $\tau(y) \neq \tau(z)$.

To deal with the incompleteness of abstract interpretation, we propose to study the following two questions for homomorphism $h : S \to \Delta$ where $h$ is either the boolean abstraction $h_{\mathbb{B}}$ or the sign abstraction $h_{\mathbb{S}}$.

**Exact Rewriting** Can we rewrite linear equation systems to $h$-exact formulas?
**Computing Abstractions** Can we can compute the abstraction $h \circ sol^S(\phi)$ exactly for a given system of homogeneous linear equations $\phi$?

Geometrically speaking, the concrete solution sets $sol^{\mathbb{R}_+}(\phi)$ and $sol^{\mathbb{R}}(\phi)$ of homogeneous linear equation systems $\phi$ are polytopes—i.e., finite intersections of half-spaces in $\mathbb{R}^{fv(\phi)}$. The problem of computing boolean abstractions or sign abstraction is thus to compute the $h_\Delta$ abstraction of a polytope given by a linear equation system.

For any $h$-exact formula $\phi$, the computation of abstractions $h \circ sol^S(\phi)$ is equivalent to the computation of $sol^\Delta(\phi)$. Since the abstract structure $\Delta$ is finite for the boolean and sign abstraction, we can compute the set of abstract solutions in at most exponential time by a naive generate and test algorithm. Finite domain constraint programming [11] can by used to avoid the naive generation of all variable assignments to $\Delta$ in many practical cases. Therefore, any algorithm for exact rewriting induces an algorithm for computing abstractions that may be feasible in practice.

### 1.2. Contributions

Our main result is the first algorithm for exact rewriting that applies to linear equation systems for the Boolean abstraction. Based on this algorithm, we present a novel algorithm for computing the sign abstraction of linear equation systems.

*Exact Rewriting for the Boolean Abstraction*. In the first step, we study exact rewriting of (homogeneous) linear equation systems for boolean abstraction. The counter example $\phi_0$, for instance, can be rewritten to $h_{\mathbb{B}}$-exact formula $y \stackrel{\circ}{=} z$. The idea is to take the system of all linear consequences over $\mathbb{R}_+$ of the linear equation system. There may be infinitely many such consequences, but all of them are linear combinations of the extreme rays of the cone $sol^{\mathbb{R}_+}(\phi_0)$. Up to normalization, there are only finitely many extreme rays, which

are known as the elementary modes of the linear equation system [12–15]. These can be computed by libraries from computational geometry [16] in at most exponential time. Nevertheless, the computation is often well-behaved in practice.

Based on the elementary modes (Folklore Theorem 2), we can rewrite any (homogeneous) linear equation system into quasi-positive and strongly-triangular linear equation system that is equivalent over $\mathbb{R}_+$ (Corollary 1), that can be computed in at most exponential time. As we prove, such systems are always $h_{\mathbb{B}}$-exact (Theorem 3). Hence, any system of linear equations can be converted in at most exponential time to some $\mathbb{R}_+$-equivalent $h_{\mathbb{B}}$-exact formula.

Note that $h_{\mathbb{B}}$-exact formulas may still not be $\mathbb{S}$-exact. A counter example is the strongly-triangular quasi-positive linear system $u + v \overset{\circ}{=} x \wedge u + v \overset{\circ}{=} y$. It is not $h_{\mathbb{S}}$-exact, since it entails $x \overset{\circ}{=} y$ over $\mathbb{R}$ but still has the abstract solution $[u/1, v/-1, x/1, y/-1]$ over $\mathbb{S}$ which maps $x$ and $y$ to distinct signs. Indeed, we don't have any idea of how to do exact rewriting for the sign abstraction. The problem is that positivity is essential for our approach, and since the addition of positive and negative numbers may have any sign, $\mathbb{S}$ fails to be an algebra, making the analogous argument as in the proof of $\mathbb{B}$-exactness fail.

*Extension to $h_{\mathbb{B}}$-Mixed Systems.* In the second step, we introduce $h_{\mathbb{B}}$-mixed systems, which by Theorem 4 generalize on systems of 1. linear equations, 2. positive polynomial equations $p \overset{\circ}{=} 0$, and 3. positive polynomial inequations $p \overset{\circ}{\neq} 0$, where $p$ is a positive polynomial without constant term. We then show our main result:

Theorem 5 (Main). Any $h_{\mathbb{B}}$-mixed systems can be converted to a $h_{\mathbb{B}}$-exact formula by converting its linear subsystem to an $h_{\mathbb{B}}$-exact formula.

The correctness of the algorithm for $h_{\mathbb{B}}$-mixed systems relies on the notion of $h_{\mathbb{B}}$-invariant formulas that we introduce. The class of $h_{\mathbb{B}}$-invariant formulas subsume systems of positive polynomial equations $p \overset{\circ}{=} 0$ and inequations $p \overset{\circ}{\neq} 0$, where $p$ is a positive polynomials without constant terms.

*Computing Sign Abstractions.* In the third step, we present an algorithm for computing the sign abstraction of (homogeneous) systems of linear equations based on exact rewriting for the boolean abstraction (Theorem 6). For this, we decompose the sign abstraction into the boolean abstraction based on a function that is definable in first-order logic. This function decomposes real numbers into their positive part $x$ and negative part $y$. At least one of these two parts must be zero, which can be expressed by the polynomial equation $x * y \overset{\circ}{=} 0$. The positivity of $x$ can be expressed by $\exists z.x \overset{\circ}{=} z * z$ and the positivity of $y$ in analogy. In this way, we can reduce the problem of computing $h_{\mathbb{S}} \circ sol^{\mathbb{R}}(\phi)$ to the problem of computing $h_{\mathbb{B}} \circ sol^{\mathbb{R}_+}(\phi')$ for some existentially quantified $h_{\mathbb{B}}$-mixed system $\phi'$ that we can make $h_{\mathbb{B}}$-exact based on our main Theorem 5.

*Application to Program Analysis.* We show how to apply the computation of the sign abstraction of linear equation systems to improve the analysis of functional programs with arithmetics. For finding program errors there it can be useful to know about the possible signs of the values of program variables. We elaborate an example in the final Section 10.

*Implementation.* We implemented the $h_{\mathbb{B}}$-exact rewriting algorithm for $h_{\mathbb{B}}$-mixed systems from the main Theorem 5 in Python. For this we used a library from computational geometry [16] for computing elementary modes. We also used finite domain constraint programming with Minizinc [17] for computing the set of boolean solutions over logical formulas. Some successful experiments are mentioned in the related work subsection below. We did not yet implemented the algorithm for computing sign abstractions, nor its application to program analysis though.

### 1.3. Related Work

We start with related work by the authors, and then move to related work by others.

*Change Prediction of Reaction Networks.* Our main Theorem 5 was recently applied to the change prediction of reaction networks in systems biology [6]. Indeed, the development of the present article was originally motivated by this application. The problem there is to compute the difference abstraction of linear equation systems, expressing the steady state semantics of chemical reaction networks. Two difference abstractions were considered, $h_{\Delta_3} : \mathbb{R}_+^2 \rightarrow \{\triangle, \triangledown, \approx\}$ and a refinement thereof $h_{\Delta_6} : \mathbb{R}_+^2 \rightarrow \{\uparrow, \downarrow, \sim, \Uparrow, \Downarrow, \approx\}$. In analogy to the approach adopted for computing sign abstractions (step three above), the algorithmic approach presented there is to decompose the difference abstractions $h_{\Delta_3}$ and $h_{\Delta_6}$ into the boolean abstraction $h_{\mathbb{B}}$ and functions that are definable in first-order logic. The elaboration of this approach, however, is quite different for reflecting the nature of the difference abstractions.

*Experimentation.* We tested our implementation of the exact rewriting algorithm for the boolean abstraction successfully for computing difference abstractions in the application of change prediction in systems biology. The experimental results are presented in [6] are generally encouraging. They show that $h_{\mathbb{B}}$-exact rewriting based on elementary modes in combination with finite domain constraint programming may indeed avoid naive generate and test in many practical examples. In some of these examples, however, the overall computation time still took some hours.

*Abstracting Reaction Networks to Boolean Networks.* Independently, the authors proposed an abstraction of chemical reaction networks to boolean networks in [18], whose precision can be improved by using the $h_{\mathbb{B}}$-exact rewriting of $h_{\mathbb{B}}$-mixed equation systems.

*Alternative Algorithm for Computing Sign Abstractions.* An alternative algorithm for computing the sign abstraction of linear equation systems (and thus also the boolean abstraction) can be obtained by John's overapproximation Theorem 1. It shows that it is sufficient to generate the finitely many abstract solutions in $\tau \in sol^{\mathbb{S}}(\phi)$, and then to check for each of them whether there exists a concrete solution $\sigma$ such that $\tau = h_{\mathbb{S}} \circ \sigma$. To perform this latter test, note that $h_\Delta(x) \overset{\circ}{=} 1$ is equivalent to the strict inequation $x > 0$ and $h_{\mathbb{S}}(x) \overset{\circ}{=} 0$ by the equation $x \overset{\circ}{=} 0$. Similarly, $h_{\mathbb{S}}(x) \overset{\circ}{=} -1$ can be defined by the strict inequation $x < 0$. Whether there exists a concrete solution $\sigma \in sol^{\mathbb{R}}(\phi)$ such $\tau = h \circ \sigma$ is thus equivalent to the satisfiability of $\phi \wedge \bigwedge_{x \in fv(\phi)} h_{\mathbb{S}}(x) \overset{\circ}{=} \tau(x)$ over $\mathbb{R}$, where $fv(\phi)$ is the set of free variables of $\phi$. The satisfiability of systems of strict linear inequations and homogeneous linear equations without constant terms over $\mathbb{R}$ are known to be decidable since at least 1926 [19]. But still, one has to generate the set of all abstract solutions $sol^{\mathbb{S}}(\phi)$. The new algorithm presented above avoids generating this set.

*Abstract Program Interpretation over Numerical Domains.* In abstract interpretation [20], nonrelational domains permit to approximate the set of values of program variables while ignoring the relationship to the values of others. Well-known nonrelational numerical domains include the interval domain [21] describing invariants of the form $\bigwedge_{i=1}^m x_i \in [r_i, r_i']$ with reals $r_i \le r_i'$ and the constant propagation domain for invariants of the form $\bigwedge_{i=1}^m x_i \overset{\circ}{=} r_i$ [22].

Abstract interpretation of relational domains may yield better approximations that with nonrelational domains, since relationships between the values of different variables can be taken into account. Well-known relational numerical domains include the polyhedral domain [4]. A polyhedron is the solution set of systems of inhomogeneous linear inequations of the form $n_1 x_1 + \ldots + n_m x_m \le r$. Alternatively, the linear equality domain [23] was considered. These are defined by system of inhomogeneous linear equations $n_1 x_1 + \ldots + n_m x_n \overset{\circ}{=} r$.

In the present paper, we study the problem of computing the sign abstraction of polytopes represented by homogeneous linear equation systems. The polytopes can be obtained by existing methods for the abstract program interpretation over the polyhedral domain. One weakness of our approach is that we study the homogeneous case only, so that we can only abstract polytope and not more general polyhedrons.

*1.4. Outline*

In Section 2, we recall preliminaries on homomorphisms between $\Sigma$-structures. In Section 3, the first-order logic is recalled. John's theorem and its relation to the soundness and completeness of abstract interpretation in the classical framework are discussed in Section 4. We discuss how to make linear equation system quasipositive and strongly triangular based on elementary modes in Section 5. These properties can be used to prove $h_\mathbb{B}$-exactness as we show in Section 6, and thus to obtain an $h_\mathbb{B}$-exact rewriting of linear equation systems. We introduce the notion of $h_\mathbb{B}$-invariance in Section 7 and apply it in Section 8 to lift the $h_\mathbb{B}$-exact rewriting algorithm from linear to $h_\mathbb{B}$-mixed systems. This allows us to define the sign abstraction of linear equation systems on Section 9. We finally apply this result in Section 10 to the sign analysis of functional programs with arithmetic.

## 2. Homomorphisms on $\Sigma$-Structures

We need some basic notation from set theory and standard notion of universal algebra such as $\Sigma$-algebras, $\Sigma$-structures, and homomorphism.

For any set $A$ and $n \in \mathbb{N}$, the set of $n$-tuples of elements in $A$ is denoted by $A^n$. For finite sets $A$ the number of elements of $A$ is denoted by $|A|$. Furthermore, for any function $f : A \to B$ we define the function $f^2 : A^2 \to B^2$ such that $f^2(a, a') = (f(a), f(a'))$ for all $a, a' \in A$.

*2.1. $\Sigma$-Algebras*

We next recall the notion of $\Sigma$-algebras. Let $\Sigma = \cup_{n \geq 0} F^{(n)} \uplus C$ be a ranked signature. We call the elements of $f \in F^{(n)}$ are called $n$-ary function symbols, even though we may also use them as $n + 1$-ary relation symbols later on when moving to $\Sigma$-structures. The elements in $c \in C$ are called the constants of $\Sigma$.

**Definition 1.** *A $\Sigma$-algebra $S = (dom(S), .^S)$ consists of a set $dom(S)$ and an interpretation $.^S$ such that $c^S \in dom(S)$ for all $c \in C$, and $f^S : dom(S)^n \to dom(S)$ for all $f \in F^{(n)}$.*

Let $\mathbb{B} = \{0, 1\}$ be the set of booleans, $\mathbb{N}$ the set of natural numbers including 0, $\mathbb{Z}$ the set of integers, $\mathbb{R}$ the set of real numbers, and $\mathbb{R}_+$ the set of positive real numbers including 0. Note that $\mathbb{B} \subseteq \mathbb{N} \subseteq \mathbb{R}_+ \subseteq \mathbb{R}$ and $\mathbb{N} \subseteq \mathbb{Z} \subseteq \mathbb{R}$. Let the addition on the reals be the binary function $+^\mathbb{R} : \mathbb{R}^2 \to \mathbb{R}$ and the multiplication the binary function $*^\mathbb{R} : \mathbb{R}^2 \to \mathbb{R}$. Let the addition on the positive real numbers $+^{\mathbb{R}_+} : \mathbb{R}_+^2 \to \mathbb{R}_+$ be equal to the restriction $+^\mathbb{R}_{|\mathbb{R}_+ \times \mathbb{R}_+}$ and the multiplication $*^{\mathbb{R}_+} : \mathbb{R}_+^2 \to \mathbb{R}_+$ be the restriction $*^\mathbb{R}_{|\mathbb{R}_+ \times \mathbb{R}_+}$.

Let $\Sigma_{bool} = \{+, *\} \cup \{0, 1\}$ be the set of boolean operators where $+$ and $*$ are binary function symbols and 0 and 1 constants. Note that constant 0 is freely overloaded with the boolean 0 and the constant 1 with the boolean 1.

**Example 1.** *The set of positive reals $\mathbb{R}_+$ can be turned into a $\Sigma_{bool}$-algebra, in which the functions symbols are interpreted as binary functions $+^{\mathbb{R}_+}$ and $*^{\mathbb{R}_+}$. The constants are interpreted by themselves $0^{\mathbb{R}_+} = 0$ and $1^{\mathbb{R}_+} = 1$.*

**Example 2.** *The set of Booleans $\mathbb{B} = \{0, 1\} \subseteq \mathbb{R}_+$ equally defines a $\Sigma_{bool}$-algebra. There, the function symbols are interpreted as a disjunction $+^\mathbb{B} = \vee^\mathbb{B}$ and conjunction $*^\mathbb{B} = \wedge^\mathbb{B}$ on Booleans. The constants are interpreted by themselves $0^\mathbb{B} = 0$ and $1^\mathbb{B} = 1$.*

*2.2. $\Sigma$-Structures*

We next recall the usual generalization of $\Sigma$-algebras to $\Sigma$-structures. The objective is to generalize from functions to relations. For this, we consider $n$-ary function symbols as $n{+}1$-ary relation symbols.

**Definition 2.** *A $\Sigma$-structure $\Delta = (dom(\Delta), .^\Delta)$ consists of a set $dom(\Delta)$ and an interpretation $.^\Delta$ such that $c^\Delta \in dom(\Delta)$ for all $c \in C$ and $f^\Delta \subseteq dom(\Delta)^{n+1}$ for all $f \in F^{(n)}$.*

Clearly, any $\Sigma$-algebra is also a $\Sigma$-structure. Note also that symbols in $F^{(0)}$ are interpreted as monadic relations, i.e., as subsets of the domain, in contrast to constants in $C$ that are interpreted as elements of the domain.

We denote the subtraction on the reals by the binary function $-^{\mathbb{R}} : \mathbb{R}^2 \to \mathbb{R}$ and the division on the reals by the ternary relation $/^{\mathbb{R}} \subseteq \mathbb{R}^2 \times \mathbb{R}$. Note that division by zero is undefined. Note also that subtraction on $\mathbb{R}_+$ would yield only a partial function.

Let $\Sigma_{arith} = \{+, *, -, /\} \cup \{0, 1\}$ be the arithmetic signature, where 0 and 1 are constants, and all other operators are binary function symbols. Again, we freely overload to constant 0 with real number 0 and the constant 1 with the real number 1.

**Example 3.** *The set of reals $\mathbb{R}$ can be turned into a $\Sigma_{arith}$-structure, with the interpretation of the binary functions symbols as the ternary relations $+^{\mathbb{R}}$, $*^{\mathbb{R}}$, $-^{\mathbb{R}}$, $/^{\mathbb{R}}$. The constants are interpreted by themselves $0^{\mathbb{R}} = 0$ and $1^{\mathbb{R}} = 1$. Note that $/^{\mathbb{R}}$ is a partial but not a total function, since division by 0 is not defined. So we must see $/^{\mathbb{R}}$ as a ternary relation, so that $\mathbb{R}$ is not a $\Sigma_{arith}$-algebra. It still is a $\Sigma_{bool}$-algebra though.*

**Example 4.** *The set of signs $\{-1, 0, 1\} \subseteq \mathbb{R}$ can be turned into a $\Sigma_{arith}$-structure $\mathbb{S} = (\{-1, 0, 1\}, .^{\mathbb{S}})$ with the interpretation $+^{\mathbb{S}}, -^{\mathbb{S}}, *^{\mathbb{S}}$ and, $/^{\mathbb{S}}$ given in Figure 1. The constants are interpreted by themselves $0^{\mathbb{S}} = 0$ and $1^{\mathbb{S}} = 1$. Note that all $+^{\mathbb{S}}$ contains $(-1, 1, -1)$, $(-1, 1, 1)$ and $(-1, 1, 0)$ meaning that the sum of a strictly negative and a strictly positive real has a sign in $-1 +^{\mathbb{S}} 1$, so it may either be strictly positive, strictly negative, or zero. So $\mathbb{S}$ is a $\Sigma_{arith}$-structure and even when restricting the signature to $\Sigma_{bool}$ it remains a $\Sigma_{bool}$-structure that is not a $\Sigma_{bool}$-algebra.*

| $d$ | $d'$ | $d +^{\mathbb{S}} d'$ | $d *^{\mathbb{S}} d'$ | $d -^{\mathbb{S}} d'$ | $d /^{\mathbb{S}} d'$ |
|---|---|---|---|---|---|
| $-1$ | $1$ | $\{-1, 0, 1\}$ | $\{-1\}$ | $\{-1\}$ | $\{-1\}$ |
| $-1$ | $0$ | $\{-1\}$ | $\{0\}$ | $\{-1\}$ | $\varnothing$ |
| $-1$ | $-1$ | $\{-1\}$ | $\{1\}$ | $\{-1, 0, 1\}$ | $\{1\}$ |

| $d$ | $d'$ | $d +^{\mathbb{S}} d'$ | $d *^{\mathbb{S}} d'$ | $d -^{\mathbb{S}} d'$ | $d /^{\mathbb{S}} d'$ |
|---|---|---|---|---|---|
| $0$ | $1$ | $\{1\}$ | $\{0\}$ | $\{-1\}$ | $\{0\}$ |
| $0$ | $0$ | $\{0\}$ | $\{0\}$ | $\{0\}$ | $\varnothing$ |
| $0$ | $-1$ | $\{-1\}$ | $\{0\}$ | $\{1\}$ | $\{0\}$ |

| $d$ | $d'$ | $d +^{\mathbb{S}} d'$ | $d *^{\mathbb{S}} d'$ | $d -^{\mathbb{S}} d'$ | $d /^{\mathbb{S}} d'$ |
|---|---|---|---|---|---|
| $1$ | $1$ | $\{1\}$ | $\{1\}$ | $\{-1, 0, 1\}$ | $\{1\}$ |
| $1$ | $0$ | $\{1\}$ | $\{0\}$ | $\{1\}$ | $\varnothing$ |
| $1$ | $-1$ | $\{-1, 0, 1\}$ | $\{-1\}$ | $\{1\}$ | $\{-1\}$ |

**Figure 1.** Evaluation in the $\Sigma_{arith}$-structure of signs $\mathbb{S}$.

*2.3. Homomorphisms*

We recall the standard notion of homomorphism for $\Sigma$-structures which can also be applied to $\Sigma$-algebras.

**Definition 3.** *A homomorphism between two $\Sigma$-structures $S$ and $\Delta$ is a function $h : dom(S) \to dom(\Delta)$ such that for $c \in C$, $n \in \mathbb{N}$, $f \in F^{(n)}$, and $s_1, \ldots, s_{n+1} \in dom(S)$:*

1.  $h(c^S) = c^\Delta$, *and*
2.  *if $(s_1, \ldots, s_{n+1}) \in f^S$ then $(h(s_1), \ldots, h(s_{n+1})) \in f^\Delta$.*

We can convert any $n + 1$-ary relation to a $n$-ary set valued functions. In this way, any $n$-function is converted to a $n$-ary set valued $n$-functions. In other words, functions of type $D^n \to D$ are converted to functions of type $D^n \to 2^D$ where $D = dom(\Delta)$. In set-valued notation, the second condition on homomorphism can then be rewritten equivalently as $h(f^S(s_1, \ldots, s_n)) \subseteq f^\Delta(h(s_1), \ldots, h(s_n))$. A homomorphism for $\Sigma$-algebras thus satisfies $h(c^S) = c^\Delta$ and for all function symbols $f \in F^{(n)}$ and $s_1, \ldots, s_n \in dom(S)$ it satisfies $h(f^S(s_1, \ldots, s_n)) = f^\Delta(h(s_1), \ldots, h(s_n))$.

The boolean abstraction is the function $h_{\mathbb{B}} : \mathbb{R}_+ \to \mathbb{B}$ with $h_{\mathbb{B}}(0) = 0$ and $h_{\mathbb{B}}(r) = 1$ if $r > 0$.

**Lemma 1.** *The boolean abstraction $h_\mathbb{B}$ is a homomorphism between $\Sigma_{bool}$-algebras.*

**Proof.** For all $r, r' \in \mathbb{R}_+$ we have:

$$h_\mathbb{B}(r +^{\mathbb{R}_+} r') = 1 \quad \Leftrightarrow \quad r +^{\mathbb{R}_+} r' \neq 0 \quad \Leftrightarrow \quad r \neq 0 \vee r' \neq 0 \quad \Leftrightarrow \quad h_\mathbb{B}(r) = 1 \vee h_\mathbb{B}(r') = 1$$
$$h_\mathbb{B}(r *^{\mathbb{R}_+} r') = 1 \quad \Leftrightarrow \quad r *^{\mathbb{R}_+} r' \neq 0 \quad \Leftrightarrow \quad r \neq 0 \wedge r' \neq 0 \quad \Leftrightarrow \quad h_\mathbb{B}(r) = 1 \wedge h_\mathbb{B}(r') = 1$$

Hence, $h_\mathbb{B}(r +^{\mathbb{R}_+} r') = h_\mathbb{B}(r) +^{\mathbb{B}} h_\mathbb{B}(r')$ and $h_\mathbb{B}(r *^{\mathbb{R}_+} r') = h_\mathbb{B}(r) *^{\mathbb{B}} h_\mathbb{B}(r')$. Finally, for both constants $c \in C$ we have that $h_\mathbb{B}(c^{\mathbb{R}_+}) = h_\mathbb{B}(c) = c = c^{\mathbb{B}}$. □

The sign abstraction is the function $h_\mathbb{S} : \mathbb{R} \to \mathbb{S}$ with $h_\mathbb{S}(0) = 0$, $h_\mathbb{S}(r) = -1$ for all strictly negative reals $r < 0$ and $h_\mathbb{S}(r) = 1$ for all strictly positive reals $r > 0$.

**Lemma 2.** *The sign abstraction $h_\mathbb{S}$ is a homomorphism between $\Sigma_{arith}$-structures.*

**Proof.** Let $r, r' \in \mathbb{R}$. For the multiplication we have $h_\mathbb{S}(r *^{\mathbb{R}} r') = h_\mathbb{S}(r) *^{\mathbb{R}} h_\mathbb{S}(r')$ and thus $h_\mathbb{S}(r *^{\mathbb{R}} r') \in \{h_\mathbb{S}(r) *^{\mathbb{R}} h_\mathbb{S}(r')\} = h_\mathbb{S}(r) *^{\mathbb{S}} h_\mathbb{S}(r')$. For the addition, we have to distinguish cases. If $r$ and $r'$ have the same sign, so $r +^{\mathbb{R}} r'$ has the same sign, so that we have $h_\mathbb{S}(r +^{\mathbb{R}} r') \in h_\mathbb{S}(r) +^{\mathbb{S}} h_\mathbb{S}(r')$. If $r > 0$ and $r' < 0$ or vice versa then we have $h_\mathbb{S}(r) +^{\mathbb{S}} h_\mathbb{S}(r') = \mathbb{S}$ so that $h_\mathbb{S}(r +^{\mathbb{R}} r') \in \mathbb{S} = h_\mathbb{S}(r) +^{\mathbb{S}} h_\mathbb{S}(r')$. The treatment of $-^{\mathbb{S}}$ and $/^{\mathbb{S}}$ is similar. For the constants, we have $h_\mathbb{S}(0^{\mathbb{R}}) = 0^{\mathbb{S}}$ and $h_\mathbb{S}(1^{\mathbb{R}}) = 1^{\mathbb{S}}$. □

## 3. First-Order Logic

We recall the syntax and semantics of first-order logic with equality. For this, we fix a countably infinite set of variables $\mathcal{V}$ that will be ranged over by $x, y, z$.

### 3.1. Expressions

Given a ranked signature with constants and function symbols $\Sigma = C \cup \bigcup_{n \geq 0} F^{(n)}$, the set of $\Sigma$-expressions contains all terms that can be constructed from constants and variables by using function symbols:

$$e_1, \ldots, e_n \in \mathcal{E}_\Sigma \quad ::= x \mid c \mid f(e_1, \ldots, e_n) \quad \text{where } c \in C, n \geq 0, f \in F^{(n)}, x \in \mathcal{V}$$

Let $fv(e)$ be the set of all variables that occur in $e$. Given a subset $V \subseteq \mathcal{V}$ let $\mathcal{E}_\Sigma(V)$ be the subset of expression $e \in \mathcal{E}_\Sigma$ with $fv(e) \subseteq V$.

The semantics of $\Sigma$-expressions is defined in Figure 2. For any $\Sigma$-structure $S$ and variable assignment $\sigma : V \to dom(S)$, any expression $e \in \mathcal{E}_\Sigma(V)$ denotes a set of values $[\![e]\!]^{\sigma,S} \subseteq dom(S)$. This set is defined recursively by set-valued interpretation of the operators of the expressions in the structure $S$. If $S$ is a $\Sigma$-algebra, then the result will always be a singleton.

$$[\![c]\!]^{\sigma,S} = \{c^S\}$$
$$[\![x]\!]^{\sigma,S} = \{\sigma(x)\}$$
$$[\![f(e_1, \ldots, e_n)]\!]^{\sigma,S} = \cup\{f^S(s_1 \ldots, s_n) \mid s_i \in [\![e_i]\!]^{\sigma,S} \text{ for } 1 \leq i \leq n\}$$

**Figure 2.** Set-valued interpretation of expressions $[\![e]\!]^{\sigma,S} \subseteq dom(S)$.

### 3.2. Logic Formulas

The set of first-order formulas is the set of terms constructed with the usual first-order connectives from equations with symbols in $\Sigma$ and variables in $\mathcal{V}$:

$$\phi \in \mathcal{F}_\Sigma \quad ::= e \overset{\circ}{=} e' \mid \exists x.\phi \mid \phi \wedge \phi \mid \neg\phi \quad \text{where } e, e' \in \mathcal{E}_\Sigma \text{ and } x \in \mathcal{V}$$

A $\Sigma$-formula $\phi \in \mathcal{F}_\Sigma$ is a term, which either is a $\Sigma$-equation $e \overset{\circ}{=} e'$ with variables in $\mathcal{V}$, an existentially quantified formula $\exists x.\phi$, a conjunction $\phi \wedge \phi'$, or a negation $\neg\phi$. A system of $\Sigma$-equations is a conjunction of equations $e_1 \overset{\circ}{=} e'_1 \wedge \ldots \wedge e_n \overset{\circ}{=} e'_n$ where $e_1, e'_1, \ldots, e_n, e'_n \in \mathcal{E}_\Sigma$.

The set of free variables $fv(\phi)$ contains all those variables of $\phi$ that occur outside the scope of any existential quantifier in $\phi$. Given a subset $V \subseteq \mathcal{V}$ we write $\mathcal{F}_\Sigma(V)$ for the subset of formulas $\phi \in \mathcal{F}_\Sigma$ such that $fv(\phi) \subseteq V$.

First-order formulas can be defined for providing the missing logical operators. Firstly, we can define disjunctions $\phi \vee \phi' =_{\text{def}} \neg(\neg\phi \wedge \neg\phi')$ and implications $\phi \rightarrow \phi' =_{\text{def}} \neg\phi \vee \phi'$, and secondly, universally quantified formulas $\forall x.\phi =_{\text{def}} \neg\exists x.\neg\phi$. Note that these formulas are not negation-free (and thus John's theorem cannot be applied to them). Third, we define the valid formula $true =_{\text{def}} \exists x.x \overset{\circ}{=} x$. Fourth, we write $\bigwedge_{i=1}^n \phi_i$ instead of $\phi_1 \wedge \ldots \wedge \phi_n$. In the case $n = 0$ the conjunction is *true*. Fifth, for any vector of variables $\mathbf{y} = (\mathbf{y}_1, \ldots, \mathbf{y}_n) \in \mathcal{V}^n$ we will write $\exists \mathbf{y}.\phi$ instead of $\exists \mathbf{y}_1 \ldots \exists \mathbf{y}_n.\phi$.

For any $V \subseteq \mathcal{V}$, the semantics of first-order formulas $\phi \in \mathcal{F}_\Sigma(V)$ for a $\Sigma$-structure $S$ and a variable assignment $\sigma : V \rightarrow dom(S)$ is the truth value $[\![\phi]\!]^{\sigma,S} \in \mathbb{B}$ defined in Figure 3.

$$[\![e \overset{\circ}{=} e']\!]^{\sigma,S} = \begin{cases} 1 & \text{if } [\![e]\!]^{\sigma,S} \cap [\![e']\!]^{\sigma,S} \neq \varnothing \\ 0 & \text{else} \end{cases} \qquad [\![\phi \wedge \phi']\!]^{\sigma,S} = [\![\phi]\!]^{\sigma,S} \wedge^{\mathbb{B}} [\![\phi']\!]^{\sigma,S}$$

$$[\![\neg\phi]\!]^{\sigma,S} = \neg^{\mathbb{B}}([\![\phi]\!]^{\sigma,S}) \qquad [\![\exists x.\phi]\!]^{\sigma,S} = \begin{cases} 1 & \text{if exists } s \in dom(S). \\ & [\![\phi]\!]^{\sigma[x/s],S} = 1 \\ 0 & \text{else} \end{cases}$$

**Figure 3.** Interpretation of formulas $\phi \in \mathcal{F}_\Sigma(V)$ as truth values $[\![\phi]\!]^{\sigma,S} \in \mathbb{B}$ over a $\Sigma$-structure $S$ given a variable assignment $\sigma : V \rightarrow dom(S)$.

Note that the equality symbol $\overset{\circ}{=}$ is interpreted as nondisjointness, i.e., an equation $e \overset{\circ}{=} e'$ is true if and only if $[\![e]\!]^{\sigma,S} \cap [\![e']\!]^{\sigma,S} \neq \varnothing$. In the case of $\Sigma$-algebras, the equality symbol $\overset{\circ}{=}$ is indeed interpreted as equality of singletons. In the case of more general $\Sigma$-structures, though, it is *not* interpreted as set equality.

The set of solutions with domain $V$ of a formula $\phi \in \mathcal{F}_\Sigma(V)$ over a $\Sigma$-algebra $S$ is:

$$sol_V^S(\phi) = \{\sigma : V \rightarrow dom(S) \mid [\![\phi]\!]^{\sigma,S} = 1\}$$

If $V = fv(\phi)$ we omit the index $V$, i.e., $sol^S(\phi) = sol_V^S(\phi)$.

Two formulas $\phi, \phi' \in \mathcal{F}_\Sigma$ are called $S$-equivalent if they have the same solution sets over $S$ on $V = fv(\phi) \cup fv(\phi')$, that is $sol_V^S(\phi) = sol_V^S(\phi')$. Note that $y \overset{\circ}{=} y$ is equivalent to $z \overset{\circ}{=} z$ and also to *true*, i.e., to $\exists x.x \overset{\circ}{=} x$.

### 3.3. Examples

Since $\mathbb{B} \subseteq \mathbb{R}_+$ we can define the boolean abstraction by a formula $y \overset{\circ}{=} h_\mathbb{B}(x)$ in $\mathcal{F}_{\Sigma_{bool}}$ over $\mathbb{R}_+$ with two variables $x, y \in \mathcal{V}$:

$$(x \overset{\circ}{=} 0 \wedge y \overset{\circ}{=} 0) \vee (\neg x \overset{\circ}{=} 0 \wedge y \overset{\circ}{=} 1)$$

Since $\mathbb{S} \subseteq \mathbb{R}$ we can define the sign abstraction by a formula $y \overset{\circ}{=} h_\mathbb{S}(x)$ in $\mathcal{F}_{\Sigma_{bool}}$ over $\mathbb{R}$ with two variables $x, y \in \mathcal{V}$:

$$(x \overset{\circ}{=} 0 \wedge y \overset{\circ}{=} 0) \vee (x > 0 \wedge y \overset{\circ}{=} 1) \vee (x < 0 \wedge y + 1 \overset{\circ}{=} 0)$$

where:

$$\begin{aligned} x \geq 0 &=_{\text{def}} \exists z.\, x \overset{\circ}{=} z * z \\ x > 0 &=_{\text{def}} x \geq 0 \wedge \neg(x \overset{\circ}{=} 0) \\ x < 0 &=_{\text{def}} \neg x \geq 0 \end{aligned}$$

These definitions illustrate that both abstraction are closely related to strict inequations $x > 0$ and $x < 0$. The boolean abstraction is concerned with strict positivity $x > 0$, while the sign abstraction is also concerned with strict negativity $x < 0$.

### 3.4. Semantic Properties of Free and Bound Variables

We need the following two standard lemmas on the role of free and bound variables for the solution sets of logic formulas. For any subset of variable assignments $R$ of type $V' \to dom(S)$ and any disjoint sets of variables $V \cap V' = \emptyset$ we define $ext^S_V(R) = \{\sigma \cup \sigma' \mid \sigma : V \to dom(S), \sigma' \in R\}$.

**Lemma 3** (Cylindrification). *If $V \cap fv(\phi) = \emptyset$ then $sol^S_{V \cup fv(\phi)}(\phi) = ext^S_V(sol^S(\phi))$.*

**Proof.** We can show that $\llbracket e \rrbracket^{\sigma,S} = \llbracket e \rrbracket^{\sigma_{|fv(e)},S}$ for all expressions $e \in \mathcal{E}_\Sigma$ with $fv(e)$ disjoint to $V$ and any variable assignment $\sigma : fv(e) \cup V \to dom(S)$ by induction on the structure of expressions. From this, we can prove for all formulas $\phi \in \mathcal{F}_\Sigma$ such that $fv(\phi)$ is disjoint from $V$ and $\sigma : fv(\phi) \cup V \to dom(S)$ that $\llbracket \phi \rrbracket^{\sigma,S} = \llbracket \phi \rrbracket^{\sigma_{|fv(\phi)},S}$ by induction on the structure of $\Sigma$-formulas. This implies the lemma. $\square$

The projection $\pi_a(f)$ of a function $f : A \to B$ is its restriction $f_{|A \setminus \{a\}}$. The projection of a set $F$ of functions $f : A \to B$ is $\pi_a(F) = \{\pi_a(f) \mid f \in F\}$.

**Lemma 4** (Quantification is projection). *$sol^S(\exists x. \phi) = \pi_x(sol^S(\phi))$.*

**Proof.** This is follows from the semantics of existential quantified formulas as follows:

$$sol^S(\exists x. \phi) = \{\sigma_{|fv(\phi) \setminus \{x\}} \mid \sigma \in sol^S(\phi)\} = \pi_x(sol^S(\phi))$$

$\square$

## 4. Abstract Interpretation

We recall the notion of $\Sigma$-abstractions and use them for abstracting sets of concrete solutions of logic formulas within the usual framework of abstract interpretation. Due to John's theorem, this abstraction can be soundly approximated by the abstract interpretation of logic formulas in the target structure of the $\Sigma$-abstraction. We will argue that John's overapproximation shows the soundness of abstract interpretation in the classical framework of Cousot & Cousot [1]. We will then introduce the notion of exactness of a logic formula with respect to a $\Sigma$-abstraction and relate it to the completeness of abstract interpretation.

### 4.1. John's Overapproximation for $\Sigma$-Abstractions

The notion of $\Sigma$-abstraction from [6] generalizes at the same time over the boolean abstraction and the sign abstraction.

**Definition 4.** *A $\Sigma$-abstraction is a homomorphism $h : S \to \Delta$ between $\Sigma$-structures such that $dom(\Delta) \subseteq dom(S)$.*

The boolean abstraction $h_{\mathbb{B}}$ is a $\Sigma_{bool}$-abstraction by Lemma 1. The sign abstraction $h_{\mathbb{S}}$ is a $\Sigma_{bool}$-abstraction by Lemma 2.

Let $h : S \to \Delta$ be a $\Sigma$-abstraction and $V \subseteq \mathcal{V}$. For any subset of assignments $R$ of type $V \to dom(S)$, we define the abstraction:

$$h \circ R = \{h \circ \sigma : V \to dom(\Delta) \mid \sigma \in R\}$$

**Theorem 1** (John's Overapproximation [6,8,9]). *For any $\Sigma$-abstraction $h : S \to \Delta$ between $\Sigma$-structures and any negation-free $\Sigma$-formula $\phi \in \mathcal{F}_\Sigma$:*

$$h \circ sol^S(\phi) \subseteq sol^\Delta(\phi)$$

John's theorem states that the abstraction with respect to $h$ of the concrete solution set of a first-order formula can be overapproximated by abstract interpretation of the formula in the target structure of $h$.

We only give a brief sketch of the full proof, which can be found in [6]. Let $V = fv(\phi)$ and $\sigma : V \to dom(S)$. For any expression $e \in \mathcal{E}_\Sigma(V)$, we can show $h(\llbracket e \rrbracket^{\sigma,S}) = \llbracket e \rrbracket^{h \circ \sigma, \Delta}$ by induction on the structure of $e$. It then follows for any negation-free formula $\phi \in \mathcal{F}_\Sigma(V)$ that $\llbracket \phi \rrbracket^{\sigma,S} \leq \llbracket \phi \rrbracket^{h \circ \sigma, \Delta}$. This is equivalent to that $\{h \circ \sigma \mid \sigma \in sol_V^S(\phi)\} \subseteq sol_V^\Delta(\phi)$ and thus $h \circ sol_V^S(\phi) \subseteq sol_V^\Delta(\phi)$. Since $V = fv(\phi)$, it follows that $h \circ sol^S(\phi) \subseteq sol^\Delta(\phi)$ as required.

### 4.2. Exactness of $\Sigma$-Formulas for $\Sigma$-Abstractions

As a new contribution, we introduce the notion of exactness of first-order formulas with respect to a $\Sigma$-abstraction.

**Definition 5** (*h*-Exactness). *Let $h : S \to \Delta$ be a $\Sigma$-abstraction and $\phi \in \mathcal{F}_\Sigma(V)$ a formula. We call $\phi$ $h$-exact with respect to $V$ if $h \circ sol_V^S(\phi) = sol_V^\Delta(\phi)$. We call $\phi$ $h$-exact if $\phi$ is $h$-exact with respect to $fv(\phi)$.*

For instance, the linear equation system $\phi$ equal to $x + y \overset{\circ}{=} x + z$ is neither $h_\mathbb{B}$-exact nor $h_\mathbb{S}$-exact. However it is equivalent to $y \overset{\circ}{=} z$ which is both $h_\mathbb{B}$-exact and $h_\mathbb{S}$-exact. To see this note that $\tau = [x/1, y/1, z/0]$ belongs to $sol^\mathbb{B}(\phi)$ but not to $h_\mathbb{B} \circ sol^{\mathbb{R}_+}(\phi)$ since $\tau(y) \neq \tau(z)$. The same assignment also belongs to $sol^\mathbb{S}(\phi)$ but not to $h_\mathbb{S} \circ sol^\mathbb{R}(\phi)$ since $\tau(y) \neq \tau(z)$.

### 4.3. Soundness and Completeness of Abstract Interpretation

John's theorem is related to the soundness of abstract interpretation and the notion of exactness to its completeness. To state the precise relationship, we need to embed our setting into the classical framework of abstract interpretation [1,10].

When considering formulas as programs, the usual framework of abstract interpretation of programs applies to the interpretation of the formulas (programs) in the target structure of the $\Sigma$-abstraction. More formally, we fix a finite subset of variables $V \subseteq \mathcal{V}$ and consider the subset of formulas as programs:

$$\mathcal{P} = \{\phi \in \mathcal{F}_\Sigma(V) \mid \phi \text{ is negation-free}\}$$

The semantics of a program $\phi \in \mathcal{P}$ over a given $\Sigma$-structure $S$ is the set of its solutions over $S$:

$$\llbracket \phi \rrbracket = sol^S(\phi)$$

The range of the semantics mapping is the space of concrete values $C = 2^{\{\sigma \mid \sigma : V \to dom(S)\}}$. Note that $(C, \subseteq, \cap, \cup)$ is a complete lattice. An abstract interpretation of a program $\phi \in \mathcal{P}$ maps $\phi$ to the set of its solutions over $\Delta$:

$$\llbracket \phi \rrbracket^\sharp = sol^\Delta(\phi)$$

The range of the abstract interpretation is the abstract domain $A = 2^{\{\tau \mid \tau : V \to dom(\Delta)\}}$. Clearly, $(A, \subseteq, \cap, \cup)$ is also a complete lattice. We define the abstraction function $\alpha_h : C \to A$ of our Galois connection such that for subsets of concrete assignments $R \subseteq C$:

$$\alpha_h(R) = h \circ R$$

**Definition 6** (Cousot & Cousot [1], Giacobazzi, Ranzato & Scozzari [10])**.** *An abstract interpretation $\llbracket . \rrbracket^\sharp : \mathcal{P} \to A$ is sound for an abstraction $\alpha : C \to A$ with respect to the program semantics $\llbracket . \rrbracket : \mathcal{P} \to C$ if for all programs $\phi \in \mathcal{P}$ it holds that $\alpha(\llbracket \phi \rrbracket) \subseteq \llbracket \phi \rrbracket^\sharp$. It is complete if all programs $\phi \in \mathcal{P}$ satisfy $\alpha(\llbracket \phi \rrbracket) = \llbracket \phi \rrbracket^\sharp$.*

John's theorem states that the abstract interpretation $\alpha_h$ of negation free-formulas $\phi \in \mathcal{P}$ over $\Delta$ is sound for the abstraction of $sol^S(\phi)$ with respect to the $\Sigma$-abstraction $h : S \to \Delta$. Furthermore, if all formulas of $\mathcal{P}$ are $h$-exact then abstract interpretation over $\Delta$ is complete for abstraction $\alpha_h$. As illustrated above, abstract interpretation over $\mathbb{B}$ fails to be complete for the abstraction $\alpha_{h_\mathbb{B}}$, and similarly, abstract interpretation over $\mathbb{S}$ fails to be complete for the abstraction $\alpha_{h_\mathbb{S}}$. Note that the completeness of abstract interpretations was largely studied in the context of program analysis (see e.g., Section 8 of [10] for an overview).

In the present article, we study the problem of exact rewriting for $h_\mathbb{B}$. The question is how to rewrite a $\Sigma_{bool}$-formula into a $h_\mathbb{B}$-exact formula that is $\mathbb{R}_+$-equivalent. Note that exact rewriting of linear equation system for $h_\mathbb{B}$ is a different problem than to decide whether abstract interpretation is complete for $\alpha_{h_\mathbb{B}}$ on linear equation systems. Still, both notions are closely related: exact rewriting can help to improve the precision of abstract interpretation just in the case where it is not already complete, i.e., maximally precise. Otherwise, exact rewriting is trivial.

In the case of the sign abstraction, we do not have any algorithmic idea of how to do exact rewriting for linear equation systems. Therefore, we study the easier problem of exact rewriting for the boolean abstraction of linear equation systems in the first place. Given an $h_\mathbb{B}$-exact formula $\phi$, we can compute the abstraction $h_\mathbb{B} \circ sol^{\mathbb{R}_+}(\phi) = sol^\mathbb{B}(\phi)$ by finite domain constraints programming. We then use exact rewriting for the boolean abstraction to compute sign abstractions of linear equation systems $h_\mathbb{S} \circ sol^\mathbb{R}(\phi)$, rather than relying on exact rewriting for the sign abstraction itself. For this, we use first-order definitions beside of finite domain constraint programming.

### 4.4. Galois Connection

We finally introduce the concretization operation that corresponds to the abstraction of the solution set of a logic formula with respect to a $\Sigma$-abstraction, and show that the pair of abstraction and concretization forms a Galois connection.

Given a $\Sigma$-abstraction $h : S \to \Delta$, and a set $R$ of variable assignments to $dom(\Delta)$, we define the left-decomposition of $R$ with respect to $h$ as the following set of variable assignments to $dom(S)$:

$$h \ominus R \quad =_{\text{def}} \{\sigma \mid h \circ \sigma \in R\}$$

So let $\alpha_h : C \to A$ be the abstraction induced by $\Sigma$-abstraction $h$. We define the corresponding concretization function $\gamma_h : A \to C$ such that for all abstract assignments $R \subseteq A$:

$$\gamma_h(R) = h \ominus R =_{\text{def}} \{\sigma \in C \mid h \circ \sigma \in R\}$$

**Lemma 5.** $(A, C, \alpha_h, \gamma_h)$ *is a Galois connection, i.e., for all $R \in C$ and $T \in A$:*

$$\alpha_h(R) \subseteq T \text{ if and only if } R \subseteq \gamma_h(T)$$

**Proof.** If $h \circ R \subseteq T$ then $h \ominus h \circ R \subseteq h \ominus T$ and since $R \subseteq h \ominus h \circ R$ we have $R \subseteq h \ominus T$. If conversely $R \subseteq h \ominus T$ then $h \circ R \subseteq h \circ h \ominus T$ and since $h \circ h \ominus T = T$ it follows that $h \circ R \subseteq T$. $\square$

## 5. Equation Systems, Positivity, and Triangularity

We study systems of $\Sigma_{bool}$-equations for positivity and triangularity. These notions will be essential for showing $\mathbb{B}$-exactness. We are not only interested in homogeneous linear equations but also in more general polynomial equations without constant term.

### 5.1. Classes of Equation Systems

Let $e_1, \ldots, e_n \in \mathcal{E}_{\Sigma_{bool}}$ be a sequence of expressions and $n \in \mathbb{N}$. If $n \neq 0$ we define $\sum_{i=1}^n e_i =_{\text{def}} e_1 + \ldots + e_n$ and $\prod_{i=1}^n e_i =_{\text{def}} e_1 * \ldots * e_n$. For $n = 0$, we define $\sum_{i=1}^n e_i = 0$ and $\prod_{i=1}^n e_i = 1$. Furthermore, for any expression $e \in \mathcal{E}_{\Sigma_{bool}}$ we define:

$$ne =_{\text{def}} \sum_{i=1}^n e \quad \text{and} \quad e^n =_{\text{def}} \prod_{i=1}^n e$$

A *polynomial (with natural coefficients)* is a $\Sigma_{bool}$-expression of the following form:

$$\sum_{j=1}^l n_j \prod_{k=1}^{i_j} x_{j,k}^{m_{j,k}}$$

where $l$ and $i_j$ are natural numbers, $x_{1,1}, \ldots, x_{l,i_l}$ variables, all $n_j \neq 0$ are natural numbers called the *coefficients*, and all $m_{j,k} \neq 0$ are natural numbers called the *exponents*. The products $\prod_{k=1}^{i_j} x_{j,k}^{m_{j,k}}$ are called the *monomials* of the polynomial.

**Definition 7.** *A polynomial $\sum_{j=1}^l n_j \prod_{k=1}^{i_j} x_{j,k}^{m_{j,k}}$ with natural coefficients $n_j \neq 0$ has* no constant term *if none of its monomials are equal to 1, i.e., $i_j \neq 0$ for all $1 \leq j \leq l$. It is* linear *if all its monomials are variables, i.e., $i_j = 1$ and $m^{j,1} = \ldots = m^{j,i_j} = 1$ for all $1 \leq j \leq l$.*

A *polynomial equation* is a $\Sigma_{bool}$-equation $p \stackrel{\circ}{=} p'$ between polynomials. A *polynomial equation system* is a system of polynomial equations.

Linear polynomials have the form $\sum_{j=1}^l n_j x_{j,1}$ where $l$ and all $n_j \neq 0$ are naturals and all $x_{j,1}$ are variables. In particular, linear polynomials do not have a constant term. Note that the constant 0 is equal to the linear polynomial with $l = 0$. A *(homogeneous) linear equation* is a polynomial equation with linear polynomials, so without constant terms. A *(homogeneous) linear equation system* is a system of linear equations.

A *(homogeneous) integer matrix equation* has the form $A\mathbf{y} \stackrel{\circ}{=} \mathbf{0}$ where $A$ is a $n \times m$ matrix of integers for some naturals $m, n$ such that $\mathbf{y} \in \mathcal{V}^m$ and $\mathbf{0} \in \{0\}^n$. Any integer matrix equation can be turned into a linear equation system with natural coefficients, by bringing the negative coefficients positively on the right-hand side. For instance, the linear integer matrix equation:

$$\begin{pmatrix} 3 & 0 \\ 2 & -5 \end{pmatrix} \begin{pmatrix} x \\ y \end{pmatrix} \stackrel{\circ}{=} \begin{pmatrix} 0 \\ 0 \end{pmatrix}$$

corresponds to the following system of linear $\Sigma_{bool}$-equations:

$$3x \stackrel{\circ}{=} 0 \wedge 2x \stackrel{\circ}{=} 5y$$

Therefore, we will sometimes confuse an integer matrix equations with the corresponding system of linear $\Sigma_{bool}$-equations. Conversely, any system of linear $\Sigma_{bool}$-equations can be converted into a integer matrix equation by moving the positive right-hand sides negatively to the left and factorizing the expressions for the different occurrences of the same variable.

### 5.2. Positivity and Triangularity

such that $\sigma(y), \sigma(y')$ We next define positivity and triangularity properties for equation systems. These are key properties to show $\mathbb{B}$-exactness of linear equation systems.

**Definition 8.** *A $\Sigma_{bool}$-equation is called* positive *if it has the form $e \stackrel{\circ}{=} 0$ and* quasi-positive *if it has the form $e \stackrel{\circ}{=} ny$, where $n \in \mathbb{N}$, $y \in \mathcal{V}$, and $e \in \mathcal{E}_{\Sigma_{bool}}$. We call a system of $\Sigma_{bool}$-equations* positive *respectively* quasi-positive *if all its equations are.*

This definition makes sense, since all constants in $\Sigma_{bool}$-expressions are positive and all operators of $\Sigma_{bool}$-expressions preserve positivity. Note also that any positive equation is quasipositive since the constant 0 is equal to the polynomial $0y$.

This above system of linear equations is quasipositive, but not positive since $5y$ appears on a right-hand side. More generally, the linear equation system for a integer matrix equation $A\mathbf{y} \stackrel{\circ}{=} \mathbf{0}$ is positive if and only if all integers in $A$ are positive, and quasipositive if each line of $A$ contains at most one negative integer.

**Definition 9.** *We call a quasipositive system of $\Sigma_{bool}$-equations* triangular *if it has the form* $\bigwedge_{l=1}^{n} e_l \stackrel{\circ}{=} n_l y_l$ *such that the variables $y_l$ are l-fresh for all $1 \leq l \leq n$, i.e., $y_l \notin fv(\bigwedge_{i=1}^{l-1} e_i \stackrel{\circ}{=} e_i')$ and if $n_l \neq 0$ then $y_l \notin fv(e_l)$. We call the quasi-positive polynomial system* strongly-triangular *if it is triangular and satisfies $n_l \neq 0$ for all $1 \leq l \leq n$.*

The above linear equation system is triangular, but not strongly triangular since the right-hand side of the first equation is 0. Consider an integer matrix equation $A\mathbf{y} \stackrel{\circ}{=} \mathbf{0}$. If $A$ is positive and triangular, then the corresponding linear equation system is positive and triangular too. For being quasipositive and strongly-triangular, the integers below the diagonal of $A$ must be negative, those on the diagonal must be strictly negative, and those on the right of the diagonal must be positive.

*5.3. Linear Equation Systems and Elementary Modes*

We next show that elementary modes [12–15] can be used to transform systems of linear equations into $\mathbb{R}_+$-equivalent systems that are *quasi-positive* and *strongly-triangular*.

We first recall the necessary definitions and folklore results on elementary modes and the double description method. We limit the presentation to equations with integer coefficients solved in $\mathbb{R}_+$, since more general definitions and results for elementary modes in $\mathbb{R}$ are not needed for this paper.

**Definition 10.** *The support of a function $\sigma : V \rightarrow \mathbb{R}$ is $supp(\sigma) = \{y \in V \mid \sigma(y) \neq 0\}$.*

**Definition 11** (Elementary Modes). *An elementary mode of an integer matrix $A \in \mathbb{Z}^{n,m}$ is a vector $\mathbf{n} \in \mathbb{N}^n$ such that for any sequence of pairwise distinct variables $\mathbf{y} \in \mathcal{V}^n$ the function $\sigma = [\mathbf{y}/\mathbf{n}]$ is a solution in $sol^{\mathbb{R}_+}(A\mathbf{y} \stackrel{\circ}{=} \mathbf{0})$ such that:*

- *$supp(\sigma)$ is minimal, i.e., there exist no $\sigma' \in sol^S(\phi)$ such that $supp(\sigma') \subsetneq supp(\sigma)$,*
- *$\sigma$ is normalized, i.e., there exist variables $y, y'$ in $\mathbf{y}$ such that $\sigma(y)$ and $\sigma(y')$ are coprimes (their greatest common divisor is 1).*

The elementary modes of a matrix $A$ are the extreme directions of the polyhedral cone $sol^{\mathbb{R}_+}(A\mathbf{y} \stackrel{\circ}{=} \mathbf{0})$. This implies that any solution of the linear system can be expressed as a weighted sum of its elementary modes, where all the weights are non negative. Due to normalization, the number of elementary modes is finite for all integer matrices.

**Theorem 2** (Folklore). *For any integer matrix $A \in \mathbb{Z}^{m,n}$ one can compute a matrix of natural numbers $E \in \mathbb{N}^{n,o}$ in at most exponential time, such that the $\Sigma_{bool}$-formulas for $A\mathbf{y} \stackrel{\circ}{=} \mathbf{0}$ and $\exists \mathbf{x}.E\mathbf{x} \stackrel{\circ}{=} \mathbf{y}$ are $\mathbb{R}_+$-equivalent for all vectors $\mathbf{y} \in \mathcal{V}^n$ and $\mathbf{x} \in \mathcal{V}^o$ of pairwise distince variables. Furthermore, the o columns of E are the elementary modes of A.*

We note that Theorem 2 can be lifted to matrices of rational numbers $\mathbb{Q}$, since any rational matrix equation $A\mathbf{y} \stackrel{\circ}{=} \mathbf{0}$ can be rewritten to a integer matrix equation with the same $\mathbb{R}_+$-solution set, by multiplying with the natural numbers in the denominators of the rational numbers. The freely available cddlib tool in the rational mode [24] inputs a matrix $A \in \mathbb{Q}^{n,m}$, and outputs the list of (integer) elementary modes of $A$. From this list, we can construct the matrix $E$ for $A$ by aligning the elementary modes of $A$ as the columns of $E$.

Note that the interface of the cddlib tool is more general, in that it applies to rational matrix inequations interpreted over the reals, rather than to rational matrix equations interpreted over the positive reals: it permits to compute the normalized extreme directions of the polyedral cone $sol^{\mathbb{R}}(B\mathbf{y} \geq \mathbf{0})$ for any rational matrix inequation $B$ over the reals. If one wants to compute the elementary modes of a rational matrix $A$ – that is the normalized extreme directions of polyhedral cones of over the positive reals $sol^{\mathbb{R}+}(A\mathbf{y} \overset{\circ}{=} \mathbf{0})$ – then one can chose $B = \begin{pmatrix} A \\ -A \\ Id \end{pmatrix}$ where $Id$ is the identity matrix with as many columns as $A$.

**Corollary 1** (Elementary Mode Rewriting). *Given a system of linear equations $\phi \in \mathcal{F}_{\Sigma_{bool}}$, one can compute in at most exponential time an $\mathbb{R}_+$-equivalent formula $emr(\phi)$ that has the form $\exists \mathbf{x}.\phi'$ where $\phi'$ is quasi-positive and strongly-triangular system of equations.*

**Proof.** Any system of linear equations $\phi$ can be converted into some integer matrix equation $A\mathbf{y} \overset{\circ}{=} \mathbf{0}$ where $\mathbf{y}$ is a vector that contains all variables in $fv(\phi)$ exactly once. Let $E$ be a matrix of elementary modes of $A$ from Theorem 2. This theorem states that $A\mathbf{y} \overset{\circ}{=} \mathbf{0}$ and thus $\phi$ is $\mathbb{R}_+$-equivalent to $\exists \mathbf{x}.E\mathbf{x} \overset{\circ}{=} \mathbf{y}$ for some vector of fresh variables $\mathbf{x}$. So let $emr(\phi)$ be $\exists \mathbf{x}.\phi'$ and $\phi'$ be $E\mathbf{x} \overset{\circ}{=} \mathbf{y}$. Since all entries of $E$ are positive, the variables in $\mathbf{y}$ are pairwise distinct, and the variables in $\mathbf{x}$ are chosen freshly, it follows that $\phi'$ is both quasi-positive and strongly-triangular. □

We have implemented the elementary mode rewriting in Python based on the cddlib tool, and plan to make our tool publically available soon. An example input is the system of linear $\Sigma_{bool}$-equations $\phi_0$ given in Figure 4. The corresponding integer matrix equation system is given there too. The elementary modes of the matrix of this system are the vectors $(1,0,1,1)$ and $(1,1,0,0)$. When putting these vectors in the columns of a new matrix, our tool returns the elementary mode rewriting $emr(\phi_0)$ in Figure 5.

$$\phi_0 =_{\text{def}} \left\{ \begin{array}{l} y_1 = y_2 + y_3 \\ \wedge \quad y_1 = y_2 + y_4 \end{array} \right. \qquad \begin{pmatrix} -1 & 1 & 1 & 0 \\ 1 & -1 & 0 & -1 \end{pmatrix} \begin{pmatrix} y_1 \\ y_2 \\ y_3 \\ y_4 \end{pmatrix} \overset{\circ}{=} \begin{pmatrix} 0 \\ 0 \\ 0 \\ 0 \end{pmatrix}$$

**Figure 4.** A linear equation system and the corresponding integer matrix equation.

$$emr(\phi_0) =_{\text{def}} \exists x_0. \exists x_1. \left\{ \begin{array}{l} x_0 + x_1 \overset{\circ}{=} y_1 \\ \wedge \quad x_1 \overset{\circ}{=} y_2 \\ \wedge \quad x_0 \overset{\circ}{=} y_3 \\ \wedge \quad x_0 \overset{\circ}{=} y_4 \end{array} \right. \qquad \begin{pmatrix} 1 & 1 \\ 0 & 1 \\ 1 & 0 \\ 1 & 0 \end{pmatrix} \begin{pmatrix} x_0 \\ x_1 \end{pmatrix} \overset{\circ}{=} \begin{pmatrix} y_1 \\ y_2 \\ y_3 \\ y_4 \end{pmatrix}$$

**Figure 5.** The elementary mode rewriting and the corresponding matrix equation.

## 6. $h_{\mathbb{B}}$-Exact Rewriting of Linear Equation Systems

Our next objective is to study the preservation of $h$-exactness by logical operators. The main difficulty of this paper is the fact that $h$-exactness is not preserved by conjunction. Nevertheless, as we will show next, it is preserved by disjunction and existential quantification.

To do so we first show that $h$-exactness is preserved when adding variables. For this, we have to assume that the $\Sigma$-abstraction $h$ is sujective, which will be the case of all $\Sigma$-abstractions of interest.

**Lemma 6** (Variable extension preserves exactness). *Let $h : S \to \Delta$ be a $\Sigma$-abstraction that is surjective and $\phi \in \mathcal{F}_\Sigma(V)$ a formula. Then the $h$-exactness of $\phi$ implies the $h$-exactness of $\phi$ with respect to $V$.*

**Proof.** This follows from that abstractions of solutions of $\phi$ can be extended arbitrarily to variables that do not appear freely in $\phi$ as stated by the following claim.

**Claim 1.** *For all* $\sigma : V \to \Delta$: $\sigma \in h \circ sol^S(\phi)$ *iff* $\sigma_{|fv(\phi)} \in h \circ sol^S(\phi)$.

For the one direction let $\sigma \in h \circ sol_V^S(\phi)$. Then, there exists $\sigma \in sol_V^S(\phi)$ such that $\sigma = h \circ \sigma$. Since $V \supseteq fv(\phi)$ it follows that $\sigma_{|fv(\phi)} \in sol^S(\phi)$. Furthermore $\sigma_{|fv(\phi)} = h \circ \sigma_{|fv(\phi)}$ and thus $\sigma_{|fv(\phi)} \in h \circ sol^S(\phi)$.

For the other direction let $\sigma_{|fv(\phi))} \in h \circ sol^S(\phi)$. Then, there exists $\sigma \in sol^S(\phi)$ such that $\sigma_{|fv(\phi)} = h \circ \sigma$. For any $y \in V \setminus fv(\phi)$ let $s_y \in dom(S)$ be such that $h(s_y) = \sigma(y)$. Such values exists since $h$ is surjective. Now define $\sigma' = \sigma[y/s_y \mid y \in V \setminus fv(\phi)]$. Since $V \supseteq fv(\phi)$ it follows that $\sigma' \in sol_V^S(\phi)$. Furthermore, $\sigma = h \circ \sigma'$, so $\sigma \in h \circ sol_V^S(\phi)$. $\square$

For the case of disjunction, we need a basic property of unions (joins) which fails for intersections (meets).

**Lemma 7** (Abstraction $\alpha_h$ preserves joins). *Let* $V$ *be a set of variables,* $R_1$ *and* $R_2$ *be subsets of assignments of type* $V \to dom(S)$ *and* $h : S \to \Delta$ *be a* $\Sigma$-*abstraction. Then:*

$$h \circ (R_1 \cup R_2) = h \circ R_1 \cup h \circ R_2$$

**Proof.** This lemma follows from the following equivalences:

$$
\begin{aligned}
\tau \in h \circ (R_1 \cup R_2) &\Leftrightarrow \exists \sigma . \sigma \in R_1 \cup R_2 \wedge \tau = h \circ \sigma \\
&\Leftrightarrow \exists \sigma . (\sigma \in R_1 \vee \sigma \in R_2) \wedge \tau = h \circ \sigma \\
&\Leftrightarrow \exists \sigma . (\sigma \in R_1 \wedge \tau = h \circ \sigma) \vee (\sigma \in R_2 \wedge \tau = h \circ \sigma) \\
&\Leftrightarrow \tau \in h \circ R_1 \vee \tau \in h \circ R_2 \\
&\Leftrightarrow \tau \in h \circ R_1 \cup h \circ R_2
\end{aligned}
$$

$\square$

**Proposition 1.** *The disjunction of h-exact formulas is h-exact.*

**Proof.** Let $\phi_1$ and $\phi_2$ be negation free formulas that are $h$-exact. Let $V = fv(\phi_1) \cup fv(\phi_2)$. Lemma 6 shows that $\phi_1$ and $\phi_2$ are also $h$-exact with respect to the extended variable set $V$, i.e., for both $i \in \{1, 2\}$:

$$h \circ sol_V^S(\phi_i) = sol_V^\Delta(\phi_i)$$

The $h$-exactness of the disjunction $\phi_1 \vee \phi_2$ can now be shown as follows:

$$
\begin{aligned}
h \circ sol^S(\phi_1 \vee \phi_2) &= h \circ (sol_V^S(\phi_1) \cup sol_V^S(\phi_2)) \\
&= h \circ sol_V^S(\phi_1) \cup h \circ sol_V^S(\phi_2) && \text{by Lemma 7} \\
&= sol_V^\Delta(\phi_1) \cup sol_V^\Delta(\phi_2) && \text{by } h\text{-exactness of } \phi_1 \text{ and } \phi_2 \text{ wrt. } V \\
&= sol^\Delta(\phi_1 \vee \phi_2)
\end{aligned}
$$

$\square$

**Lemma 8** (Projection commutes with abstraction). *For any* $\Sigma$-*abstraction* $h : S \to \Delta$, *subset* $R$ *of assignments of type* $V \to S$, *and variable* $x \in \mathcal{V}$: $h \circ \pi_x(R) = \pi_x(h \circ R)$.

**Proof.** For all $\sigma : V \to dom(S)$ we have $h \circ \pi_x(\sigma) = h \circ \sigma_{|V \setminus \{x\}} = (h \circ \sigma)_{|V \setminus \{x\}} = \pi_x(h \circ \sigma)$. $\square$

**Proposition 2** (Quantification preserves exactness). *For any surjective* $\Sigma$-*abstraction* $h : S \to \Delta$ *and formula* $\exists x . \phi \in \mathcal{F}_\Sigma$, *if* $\phi$ *is h-exact then* $\exists x . \phi$ *is h-exact.*

**Proof.** Let $\phi$ be $h$-exact. By definition $\phi$ is $h$-exact with respect to $V = fv(\phi)$. Since $h$ is assumed to be surjective, Lemma 6 implies that $\phi$ is $h$-exact with respect to $V \cup \{x\}$ (independently of whether $x$ occurs freely in $\phi$ or not). Hence:

$$
\begin{aligned}
h(sol^S(\exists x.\phi)) &= h(\pi_x(sol^S(\phi))) && \text{by Lemma 4} \\
&= \pi_x(h(sol^S(\phi))) && \text{by Lemma 8} \\
&= \pi_x(sol^\Delta(\phi)) && \text{since } \phi \text{ is } h\text{-exact} \\
&= sol^\Delta(\exists x.\phi) && \text{by Lemma 4}
\end{aligned}
$$

$\square$

We next study the $h$-exactness for strongly-triangular systems of $\Sigma_{bool}$-equations, under the condition that $h$ is an abstraction between $\Sigma_{bool}$-algebras with unique division (see Definition 12).

**Lemma 9** (Singleton property). *If $S$ is a $\Sigma$-algebra, $e \in \mathcal{E}_\Sigma(V)$, and $\sigma : V \to S$ a variable assignment, then the set $[\![e]\!]^{\sigma,S}$ is a singleton.*

**Proof.** By induction on the structure of expressions $e \in \mathcal{E}_\Sigma(V)$:
**Case** of constants $c \in C$. The set $[\![c]\!]^{\sigma,S} = \{c^S\}$ is a singleton.
**Case** of variables $x \in V$. The set $[\![x]\!]^{\sigma,S} = \{\sigma(x)\}$ is a singleton.
**Case** $f(e_1,\ldots,e_n)$ where $e_i \in \mathcal{E}_\Sigma(V)$ and $f \in F^{(n)}$.

$$
[\![f(e_1,\ldots,e_n)]\!]^{\sigma,S} = \{f^S(s_1,\ldots,s_n) \mid s_i \in [\![e_i]\!]^{\sigma,S}\}
$$

This set is a singleton since $[\![e_i]\!]^{\sigma,S}$ are singletons by induction hypothesis, meaning that $f^S([\![e_1]\!]^{\sigma,S},\ldots,[\![e_n]\!]^{\sigma,S})$ is also a singleton since $S$ is a $\Sigma$-algebra. $\square$

A $\Sigma$-algebra is a $\Sigma$-structure with the singleton property. Let *ele* be the function that maps any singleton to the element that it contains.

**Definition 12.** *We say that a $\Sigma_{bool}$-structure $S$ has unique division if it satisfies the first-order formula $\forall x.\exists^{=1}y. \; ny \overset{\circ}{=} x$ for all nonzero natural numbers $n \in \mathbb{N}$.*

Clearly, the $\Sigma_{bool}$-structures $\mathbb{R}_+$, $\mathbb{B}$, and $\mathbb{S}$ have unique division. Note, however, that $\mathbb{S}$ is not a $\Sigma_{bool}$-algebra, so that the following two Propositions 3 and 4 cannot be applied to $\mathbb{S}$ instead of $\mathbb{B}$.

For any element $s$ of the domain of a $\Sigma_{bool}$-structure $S$ with unique division and any nonzero natural number $n \in \mathbb{N}$, we denote by $\frac{s}{n}$ the unique element of $\{\sigma(y) \mid \sigma \in sol^S(ny \overset{\circ}{=} z), \sigma(z) = s\}$.

**Lemma 10.** *Let $\phi \in \mathcal{F}_{\Sigma_{bool}}$ be a formula and $S$ a $\Sigma_{bool}$-algebra with unique division. For nonzero natural number $n$, variable $y \notin fv(\phi)$, and expression $e \in \mathcal{E}_\Sigma(fv(\phi))$:*

$$
sol^S(\phi \wedge ny \overset{\circ}{=} e) = \left\{ \sigma[y/\frac{ele([\![e]\!]^{\sigma,S})}{n}] \;\middle|\; \sigma \in sol^S(\phi) \right\}
$$

**Proof.** We fix some $\sigma : fv(\phi) \to dom(S)$ arbitrarily. Since $S$ is a $\Sigma_{bool}$-algebra, $[\![e]\!]^{\sigma,S}$ is a singleton and $fv(e) \subseteq V(\phi)$, $ele([\![e]\!]^{\sigma,S})$ is defined uniquely. Furthermore $S$ has unique division, so that $\frac{ele([\![e]\!]^{\sigma,S})}{n}$ is a well-defined element of $dom(S)$. Therefore and since $y \notin fv(\phi)$, $\sigma[y/\frac{ele([\![e]\!]^{\sigma,S})}{n}]$ is the unique solution of the equation $ny \overset{\circ}{=} e$ that extends on $\sigma$.

Firstly, we prove the inclusion "$\supseteq$". Let $\sigma \in sol^S(\phi)$, $y \notin fv(\phi)$, and $\sigma[y/\frac{ele([\![e]\!]^{\sigma,S})}{n}]$ is a solution of $ny \overset{\circ}{=} e$, it follows that $\sigma[y/\frac{ele([\![e]\!]^{\sigma,S})}{n}]$ is a solution of $\phi \wedge ny \overset{\circ}{=} e$.

Secondly, we prove the inverse inclusion "⊆". Let $\sigma \in sol^S(\phi \wedge ny \overset{\circ}{=} e)$. Since $\sigma[y / \frac{ele(\llbracket e \rrbracket^{\sigma,S})}{n}]$ is the unique solution of the equation $ny \overset{\circ}{=} e$ that extends on $\sigma' = \sigma_{|fv(\phi)}$ it follows that $\sigma(y) = \frac{ele(\llbracket e \rrbracket^{\sigma,S})}{n}$ so that $\sigma = \sigma'[y / \frac{ele(\llbracket e \rrbracket^{\sigma,S})}{n}]$ while $\sigma' \in sol^S(\phi)$. □

**Proposition 3.** *Let $\phi \in \mathcal{F}_{\Sigma_{bool}}(V)$ a formula, $n \neq 0$ a natural number, $e \in \mathcal{E}_{\Sigma_{bool}}(V)$ an expression, $y \notin V$, and $h : S \to \Delta$ a $\Sigma_{bool}$-abstraction between $\Sigma_{bool}$-algebras with unique division. Under these conditions, if $\phi$ is h-exact then $\phi \wedge e \overset{\circ}{=} ny$ is h-exact.*

**Proof.** Let $e \in \mathcal{E}_{\Sigma_{bool}}(V)$ an expression. □

**Claim 2.** *For any $\sigma : V \to \mathbb{R}_+$: $h(ele(\llbracket e \rrbracket^{\sigma,S})) = ele(\llbracket e \rrbracket^{h \circ \sigma, \Delta})$.*

This can be seen as follows. For any $\sigma : V \to S$ Theorem 1 on homomorphism yields $h(\llbracket e \rrbracket^{\sigma,S}) \subseteq \llbracket e \rrbracket^{h \circ \sigma, \Delta}$. Since $S$ and $\Delta$ are both $\Sigma$-algebras, the sets $\llbracket e \rrbracket^{\sigma,S}$ and $\llbracket e \rrbracket^{h \circ \sigma, \Delta}$ are both singletons by Lemma 9, so that $h(ele(\llbracket e \rrbracket^{\sigma,S})) = ele(\llbracket e \rrbracket^{h \circ \sigma, \Delta})$.

**Claim 3.** *For any $s \in dom(S)$ and $n \neq 0$ a natural number: $h(\frac{s}{n}) = \frac{h(s)}{n}$.*

Since $S$ is assumed to have unique division $s' = \frac{s}{n}$ is well-defined as the unique element of $dom(S)$ such that $\underbrace{s' +^S \ldots +^S s'}_{n} = s$. Hence, $h(\underbrace{s' +^S \ldots +^S s'}_{n}) = h(s)$ and since $h$ is a homomorphism, it follows that $\underbrace{h(s') +^\Delta \ldots +^\Delta h(s')}_{n} = h(s)$. Since $\Delta$ is assumed to have unique division, this implies that $h(s') = \frac{h(s)}{n}$.

The Proposition can now be shown based on these two claims. Let $\phi$ be h-exact, $y \notin V$, and $fv(e) \subseteq V$. We have to show that $\phi \wedge ny \overset{\circ}{=} e$ is h-exact too:

$$
\begin{aligned}
h \circ sol^S(\phi \wedge e \overset{\circ}{=} ny) &= h \circ \{\sigma[y / \tfrac{ele(\llbracket e \rrbracket^{\sigma,S})}{n}] \mid \sigma \in sol^S(\phi)\} && \text{by Lemma 10} \\
&= \{(h \circ \sigma)[y / h(\tfrac{ele(\llbracket e \rrbracket^{\sigma,S})}{n})] \mid \sigma \in sol^S(\phi)\} && \text{elementary} \\
&= \{\sigma[y / h(\tfrac{ele(\llbracket e \rrbracket^{\sigma,S})}{n})] \mid \sigma \in sol^\Delta(\phi)\} && \text{h-exactness of } \phi \\
&= \{\sigma[y / \tfrac{h(ele(\llbracket e \rrbracket^{\sigma,S}))}{n}] \mid \sigma \in sol^\Delta(\phi)\} && \text{by Claim 3} \\
&= \{\sigma[y / \tfrac{ele(\llbracket e \rrbracket^{h \circ \sigma, \Delta})}{n}] \mid \sigma \in sol^\Delta(\phi)\} && \text{by Claim 2} \\
&= sol^\Delta(\phi \wedge e \overset{\circ}{=} ny) && \text{by Lemma 10}
\end{aligned}
$$

**Proposition 4.** *Let $h : S \to \Delta$ be a $\Sigma_{bool}$-abstraction between algebras with unique division. Then any strongly-triangular system of $\Sigma_{bool}$-equations is h-exact.*

**Proof.** Any strongly-triangular system of equations has the form $\bigwedge_{i=1}^n e_i \overset{\circ}{=} n_i y_i$ where $n$ and $n_i \neq 0$ are naturals and $y_i$ is $i$-fresh for all $1 \leq i \leq n$. The proof is by induction on $n$. In the case $n = 0$, the conjunction is equal to *true* which is h-exact since $h(sol^S(true)) = sol^\Delta(true)$. In the case $n > 0$, we have by induction hypothesis that $\bigwedge_{j=1}^{i-1} e_j \overset{\circ}{=} n_j y_j$ is h-exact. Since $n_i \neq 0$ it follows from Proposition 3 that that $e_i \overset{\circ}{=} n_i y_i \wedge \bigwedge_{j=1}^{i-1} e_j \overset{\circ}{=} n_j y_j$ is h-exact. □

We notice that Proposition 4 remains true for triangular systems that are not strongly-triangular. This follows from results that we can only present in the next section (Theorem 4 and Proposition 5), since they require an additional argument.

**Theorem 3** ($h_\mathbb{B}$-Exactness)**.** *Quasi-positive strongly-triangular polynomial systems are $h_\mathbb{B}$-exact.*

**Proof.** The $\Sigma_{bool}$-algebras $\mathbb{R}_+$ and $\mathbb{B}$ have unique division, so we can apply Proposition 4 for proving the theorem. $\square$

We note that the analogous statement for $\mathbb{S}$ instead of $\mathbb{B}$ fails, even though $\mathbb{S}$ has unique division. The problem is that $\mathbb{S}$ is not a $\Sigma_{bool}$-algebra. As a counter-example, reconsider the strongly-triangular system of quasi-positive system equations:

$$u + v \overset{\circ}{=} x \wedge u + v \overset{\circ}{=} y$$

This system implies $x \overset{\circ}{=} y$ over $\mathbb{R}$ but accept the abstract solution $[u/1, v/-1, x/1, y/-1]$ mapping $x$ and $y$ to distinct signs, so it is not $h_{\mathbb{S}}$-exact. Nevertheless, it is $h_{\mathbb{B}}$-exact by Theorem 3.

**Corollary 2** ($h_{\mathbb{B}}$-exact rewriting of linear equation systems). *For any linear $\Sigma_{bool}$-equations $\phi$ the elementary mode rewriting $emr(\phi) \in \mathcal{F}_{\Sigma_{bool}}$ is $\mathbb{R}_+$-equivalent, $h_{\mathbb{B}}$-exact, and can be computed in at most exponential time from $\phi$.*

**Proof.** The elementary modes rewriting Corollary 1 shows that any linear $\Sigma_{bool}$-equation system $\phi$ is $\mathbb{R}_+$-equivalent a formula $emr(\phi)$ of the form $\exists \mathbf{z}.\phi'$ such that $\phi'$ is a quasi-positive strongly-triangular linear equation system. Theorem 3 shows that any quasi-positive strongly-triangular linear equation system is $h_{\mathbb{B}}$-exact, so is $\phi'$. Existential quantification preserves $h_{\mathbb{B}}$-exactness by Proposition 2, so $emr(\phi)$ is $h_{\mathbb{B}}$-exact too. $\square$

This $h_{\mathbb{B}}$-exact rewriting permits us to compute the boolean abstraction of any system of linear $\Sigma_{bool}$-equations by computing the $\mathbb{B}$-solutions of the $\mathbb{R}_+$-equivalent $h_{\mathbb{B}}$-exact formula. The latter can be done by finite domain constraint programming.

Our objective to find an algorithm for computing the sign abstraction of a system of linear $\Sigma_{bool}$-equations remains open. We finally approach it in Section 9. While the idea is to use the $h_{\mathbb{B}}$-exact rewriting algorithm, we first need to generalize it from linear systems to mixed systems. This is done in Section 8. The generalization relies on the notion of $h_{\mathbb{B}}$-invariance, which we discuss next in Section 7.

## 7. Invariance

A problem that we need to overcome is that conjunctions of two $h$-exact formulas may not be $h$-exact. The situation changes when assuming the following notion of $h$-invariance for at least one of the two formulas.

**Definition 13** (Invariance). *Let $h : S \rightarrow \Delta$ be a $\Sigma$-abstraction and $V \subseteq \mathcal{V}$ a subset of variables. We call a subset $R$ of variable assignments of type $V \rightarrow dom(S)$ $h$-invariant iff:*

$$\forall \sigma, \sigma' : V \rightarrow dom(S). \ (\sigma \in R \wedge h \circ \sigma = h \circ \sigma' \implies \sigma' \in R).$$

*We call a $\Sigma$-formula $\phi$ $h$-invariant if its solution set $sol^S(\phi)$ is.*

The relevance of the notion of invariance for exactness of conjunctions—that we will formalize in Proposition 5—is due to the the following lemma:

**Lemma 11.** *If either $R_1$ or $R_2$ are $h$-invariant then: $h \circ (R_1 \cap R_2) = h \circ R_1 \cap h \circ R_2$.*

**Proof.** The one inclusion is straightforward without invariance:

$$
\begin{aligned}
h \circ (R_1 \cap R_2) &= \{h \circ \sigma \mid \sigma \in R_1, \ \sigma \in R_2\} \\
&\subseteq \{h \circ \sigma \mid \sigma \in R_1\} \cap \{h \circ \sigma \mid \sigma \in R_2\} \\
&= h \circ R_1 \cap h \circ R_2
\end{aligned}
$$

For the other inclusion, we can assume without loss of generality that $R_1$ is $h$-invariant. So let $\tau \in h \circ R_1 \cap h \circ R_2$. Then, there exist $\sigma_1 \in R_1$ and $\sigma_2 \in R_2$ such that $\tau = h \circ \sigma_1 = h \circ \sigma_2$. By $h$-invariance of $R_1$ it follows that $\sigma_1 \in R_2$. So $\sigma_1 \in R_1 \cap R_2$, and hence, $\tau \in h \circ (R_1 \cap R_2)$. $\square$

We can now present the algebraic characterization of $h$-invariance based on the concretization function $\gamma_h$ of the Galois connection of $h$. Recall that $R \subseteq h \ominus (h \circ R)$ for all subsets of concrete variable assignments $R$. The inverse inclusion characterizes the $h$-invariance of $R$.

**Lemma 12** (Algebraic characterization). *Let $h : S \to \Delta$ be a $\Sigma$-abstraction. A subset $R$ of concrete variable assignment $V \to dom(S)$ is $h$-invariant for $h$ iff $h \ominus (h \circ R) \subseteq R$.*

**Proof.** "$\Rightarrow$". Let $R$ be $h$-invariant and $\sigma \in h \ominus (h \circ R)$. Then, there exists $\sigma' \in R$ such that $h \circ \sigma = h \circ \sigma'$. The $h$-invariance of $R$ thus implies that $\sigma \in R$.
"$\Leftarrow$". Suppose that $h \ominus (h \circ R) \subseteq R$. Let $\sigma, \sigma' : V \to dom(S)$ such that $h \circ \sigma = h \circ \sigma'$ and $\sigma \in R$. We have to show that $\sigma' \in R$. From $h \circ \sigma = h \circ \sigma'$ and $\sigma \in R$ it follows that $\sigma' \in h \ominus (h \circ R)$ and thus $\sigma' \in R$ as required. $\square$

**Lemma 13** (Variable extension preserves invariance). *Let $h$ be a surjective abstraction and $R$ a subset of functions of type $V' \to dom(S)$ and $V$ a subset of variables disjoint from $V'$. If $R$ is $h$-invariant then $ext_V^S(R)$ is $h$-invariant too.*

**Proof.** This follows straightforwardly from the characterization of $h$-invariance in Lemma 12 and the following two claims:

**Claim 4.** *If $h$ is surjective then $h \circ ext_V^S(R) = ext_V^\Delta(h \circ R)$.*

This follows from $h \circ ext_V^S(R) = \{h \circ \sigma \mid \sigma \in ext_V^S(R)\} = ext_V^\Delta(\{h \circ \sigma' \mid \sigma' \in R\})$ where we use the surjectivity of $h$ in the last step.

**Claim 5.** $h \ominus ext_V^\Delta(R') = ext_V^S(h \ominus R')$ *for any subset $R'$ of functions of type $V' \to dom(\Delta)$.*

$$
\begin{aligned}
h \ominus ext_V^\Delta(R') &= \{\sigma : V \cup V' \to dom(S) \mid h \circ \sigma \in ext_V^\Delta(R')\} \\
&= \{\sigma : V \cup V' \to dom(S) \mid h \circ \sigma_{|V'} \in R'\} \\
&= ext_V^S(\{\sigma' : V' \to dom(S) \mid h \circ \sigma' \in R'\} \\
&= ext_V^S(h \ominus R')
\end{aligned}
$$

$\square$

**Lemma 14.** *Let $h : S \to \Delta$ be a surjective $\Sigma$-abstraction, $\phi$ be a $\Sigma$-formula, and $V \supseteq fv(\phi)$. Then, the $h$-invariance of $\phi$ implies the $h$-invariance of $sol_V^S(\phi)$.*

**Proof.** This follows from the cylindrification Lemma 3 and that extension preserves $h$-invariance as shown in Lemma 13. $\square$

**Proposition 5** (Exactness is preserved by conjunction when assuming invariance). *Let $h$ be a surjective $\Sigma$-abstraction. If $\phi_1$ and $\phi_2$ are $h$-exact $\Sigma$-formulas and $\phi_1$ or $\phi_2$ are $h$-invariant then the conjunction $\phi_1 \wedge \phi_2$ is $h$-exact.*

**Proof.** Let $\phi_1$ and $\phi_2$ be $h$-exact $\Sigma$-formulas. We assume without loss of generality that $\phi_1$ is $h$-invariant. Let $V = fv(\phi_1 \wedge \phi_2)$. Since $fv(\phi_2) \subseteq V$ the set $sol_V^S(\phi_2)$ is $h$-invariant too by Lemma 14. We can now show that $\phi_1 \wedge \phi_2$ is $h$-exact as follows:

$$
\begin{aligned}
h \circ sol^S(\phi_1 \wedge \phi_2) &= h \circ (sol_V^S(\phi_1) \cap sol_V^S(\phi_2)) \\
&= h \circ sol_V^S(\phi_1) \cap h \circ sol_V^S(\phi_2) \quad \text{by Lemma 11} \\
&= sol_V^\Delta(\phi_1) \cap sol_V^\Delta(\phi_2) \quad\quad\quad \text{by } h\text{-exactness of } \phi_1 \text{ and } \phi_2 \text{ wrt } V \\
&= sol^\Delta(\phi_1 \wedge \phi_2)
\end{aligned}
$$

$\square$

Our next objective is to show that $h$-invariant formulas are closed under conjunction, disjunction, and existential quantification. The two former closure properties rely on the following two algebraic properties of abstraction decomposition.

**Lemma 15** (Concretization $\gamma_h$ preserves join and meet). *For any $\Sigma$-abstraction $h : S \to \Delta$, any subsets of assignments of type $V \to dom(S)$ $R_1$ and $R_2$ and $V$ a subset of variables:*
- $h \ominus (R_1 \cap R_2) = h \ominus R_1 \cap h \ominus R_2.$
- $h \ominus (R_1 \cup R_2) = h \ominus R_1 \cup h \ominus R_2.$

For general Galois connections, concretization is well-known to preserve joins but may not preserve meets. Still, meets are preserved for any Galois connections where the the concrete and abstract domains $C$ and $A$ are powersets as in our setting, so that joins are unions and meets intersections.

**Proof.** The case of unions follows straightforwardly from the definitions:

$$
\begin{aligned}
h \ominus (R_1 \cup R_2) &= \{\sigma \mid h \circ \sigma \in R_1 \cup R_2\} \\
&= \{\sigma \mid h \circ \sigma \in R_1 \vee h \circ \sigma \in R_2\} \\
&= \{\sigma \mid h \circ \sigma \in R_1\} \cup \{\sigma \mid h \circ \sigma \in R_2\} \\
&= h \ominus R_1 \cup h \ominus R_2
\end{aligned}
$$

The case of intersection is symmetric:

$$
\begin{aligned}
h \ominus (R_1 \cap R_2) &= \{\sigma \mid h \circ \sigma \in R_1 \cap R_2\} \\
&= \{\sigma \mid h \circ \sigma \in R_1 \wedge h \circ \sigma \in R_2\} \\
&= \{\sigma \mid h \circ \sigma \in R_1\} \cap \{\sigma \mid h \circ \sigma \in R_2\} \\
&= h \ominus R_1 \cap h \ominus R_2
\end{aligned}
$$

$\square$

**Lemma 16** (Intersection and union preserve invariance). *Let $h : S \to \Delta$ be a $\Sigma$-abstraction. Then, the intersection and union of any two $h$-invariant subsets $R_1$ and $R_2$ of variables assignments of type $V \to dom(S)$ is $h$-invariant.*

**Proof.** This follows from the algebraic characterization Lemma 12 for invariance, in combination with the algebraic properties of composition and decomposition given in Lemmas 7, 11, and 15. $\square$

**Lemma 17** (Concretization $\gamma_h$ commutes with projection). $h \ominus \pi_x(R) = \pi_x(h \ominus R).$

**Proof.** For all $\sigma : V \to dom(\Delta)$ we have $h \ominus \pi_x(\sigma) = h \ominus \sigma_{|V \setminus \{x\}} = (h \ominus \sigma)_{|V \setminus \{x\}} = \pi_x(h \ominus \sigma).$ $\square$

**Proposition 6** (Invariance is preserved by conjunction, disjunction, and quantification)**.** *If h is a surjective abstraction, then the class of h-invariant FO-formulas is closed under conjunction, disjunction, and existential quantification.*

**Proof.** Let $h : S \to \Delta$ be a $\Sigma$-abstraction.
**Case** of conjunction: Let $\phi_1$ and $\phi_2$ be $h$-invariant and $V = fv(\phi_1 \wedge \phi_2)$. By Lemma 14 the sets $sol_V^S(\phi_1)$ and $sol_V^S(\phi_2)$ are both $h$-invariant, and so by Lemma 16 is their intersection. Hence:

$$
\begin{aligned}
h \ominus (h \circ sol^S(\phi_1 \wedge \phi_2)) & \\
= \quad & h \ominus (h \circ (sol_V^S(\phi_1) \cap sol_V^S(\phi_2))) \\
\subseteq \quad & sol_V^S(\phi_1) \cap sol_V^S(\phi_2) \qquad \text{by } h\text{-invariance and Lemma 12} \\
= \quad & sol^S(\phi_1 \wedge \phi_2)
\end{aligned}
$$

By Lemma 12 in the other direction, this implies that $\phi_1 \wedge \phi_2$ is $h$-invariant.
**Case** of disjunction: Analogous to the case of conjunction.
**Case** of existential quantification:

$$
\begin{aligned}
h \ominus (h \circ sol^S(\exists x.\phi_1)) & \\
= \quad & h \ominus (h \circ \pi_x(sol^S(\phi_1))) & \text{by Lemma 4} \\
= \quad & h \ominus (\pi_x(h \circ sol^S(\phi_1))) & \text{by Lemma 8} \\
= \quad & \pi_x(h \ominus (h \circ sol^S(\phi_1))) & \text{by Lemma 17} \\
\subseteq \quad & \pi_x(sol^S(\phi_1)) & \text{by } h\text{-invariance of } \phi_1 \text{ and Lemma 12} \\
= \quad & sol^S(\exists x.\phi_1) & \text{by Lemma 4}
\end{aligned}
$$

By Lemma 12, this implies that $\exists x.\phi_1$ is $h$-invariant. $\square$

We do not know whether negation preserves $h$-invariance in general, but for finite $\Delta$ it can be shown that if $\phi$ is $h$-exact and $h$-invariant, then $\neg \phi$ is $h$-exact and $h$-invariant too.

**Proposition 7.** *Let $h$ be a surjective $\Sigma$-abstraction. Then, the class of h-exact and h-invariant $\Sigma$-formulas is closed under conjunction, disjunction, and existential quantification.*

**Proof.** Closure under conjunction follows from Propositions 5 and 6, closure under disjunction from Propositions 1 and 6, and closure under existential quantification by Propositions 2 and 6. $\square$

**Theorem 4** ($h_{\mathbb{B}}$-invariance and $h_{\mathbb{B}}$-exactness of polynomial equations)**.** *Any positive polynomial equation $p \overset{\circ}{=} 0$ such that $p$ has no constant term is $h_{\mathbb{B}}$-exact and $h_{\mathbb{B}}$-invariant.*

**Proof.** Consider a positive polynomial equation $p \overset{\circ}{=} 0$ such that $p$ has no constant term and only positive coefficients. Thus, $p$ has the form $\sum_{j=1}^{l} n_j \prod_{k=1}^{i_j} x_{j,k}^{m_{j,k}} \overset{\circ}{=} 0$ where $l \geq 0$, and $n_j, i_j, m_{j,k} > 0$.

**Claim 6.** *For both algebras $S \in \{\mathbb{B}, \mathbb{R}_+\}$: $sol^S(p \overset{\circ}{=} 0) = sol^S(\bigwedge_{j=1}^{l} \bigvee_{k=1}^{i_j} x_{j,k} \overset{\circ}{=} 0)$.*

The polynomial has a value of zero if and only if all its monomials do, that is: $\prod_{k=1}^{i_j} x_{j,k}^{m_{jk}} = 0$ for all $1 \leq j \leq l$. Since constant terms are ruled out, we have $i_j \neq 0$. Furthermore, we assumed for all polynomials that $m_{j,k} \neq 0$. So for all $1 \leq j \leq l$ there must exist $1 \leq k \leq i_j$ such that $x_{j,k} = 0$.

**Claim 7.** *The equation $x \overset{\circ}{=} 0$ is $h_{\mathbb{B}}$-exact and $h_{\mathbb{B}}$-invariant.*

This proof of this claim is straightforward from the definitions.

With these two claims, we are now in the position to prove the Theorem 4. Since the class of $h_\mathbb{B}$-exact and $h_\mathbb{B}$-invariant formulas is closed under conjunction and disjunction by Proposition 7, it follows from by Claim 7 that $\wedge_{j=1}^{l} \vee_{k=1}^{i_j} x_{j,k} \overset{\circ}{=} 0$ is both $h_\mathbb{B}$-exact and $h_\mathbb{B}$-invariant. Since this formula is equivalent over $\mathbb{R}_+$ to the polynomial equation by Claim 6, the $h_\mathbb{B}$-invariance carries over to $p \overset{\circ}{=} 0$. The $h_\mathbb{B}$-exactness also carries over based on the equivalence for both structures $\mathbb{R}_+$ and $\mathbb{B}$:

$$
\begin{aligned}
h_\mathbb{B} \circ sol^{\mathbb{R}_+}(p \overset{\circ}{=} 0) &= h_\mathbb{B} \circ sol_V^{\mathbb{R}_+}(\wedge_{j=1}^{l} \vee_{k=1}^{i_j} x_{j,k} \overset{\circ}{=} 0) && \text{by Claim 6 for } \mathbb{R}_+ \\
&= sol^{\mathbb{B}}(\wedge_{j=1}^{l} \vee_{k=1}^{i_j} x_{j,k} \overset{\circ}{=} 0) && \text{by } h_\mathbb{B} \text{ exactness} \\
&= sol^{\mathbb{B}}(p \overset{\circ}{=} 0) && \text{by Claim 6 for } \mathbb{B}.
\end{aligned}
$$

□

### 8. $h_\mathbb{B}$-Exact Rewriting of $h_\mathbb{B}$-Mixed Systems

In this section, we lift our main result to $h_\mathbb{B}$-mixed system, presenting a rewrite algorithm that makes any $h_\mathbb{B}$-mixed system $h_\mathbb{B}$-exact.

**Definition 14.** *A $h_\mathbb{B}$-mixed system is a formula in $\mathcal{F}_{\Sigma_{bool}}$ of the form $\exists \mathbf{z}.\ \phi \wedge \phi'$ where $\phi$ is a system of linear $\Sigma_{bool}$-equations and $\phi'$ a $h_\mathbb{B}$-invariant and $h_\mathbb{B}$-exact first-order formula.*

Note that linear equation systems $A\mathbf{y} \overset{\circ}{=} \mathbf{0}$, with $A$ an integer matrix and $\mathbf{y}$ a sequence of pairwise distinct variables, need not to be $h_\mathbb{B}$-exact, if $A$ is not positive. However, as shown by the elementary mode rewriting Corollary 1 any linear equation systems is $\mathbb{R}_+$-equivalent to some quasipositive strongly-triangular linear system, that is $h_\mathbb{B}$-exact by Theorem 3.

Our next objective is to rewrite formulas to reduce the overapproximation coming with the abstract interpretation over the Booleans by John's theorem. The idea is to make a linear equation system $h_\mathbb{B}$-exact that are used as subformulas as for instance of $h_\mathbb{B}$-mixed systems.

We recall from Corollary 1 that the elementary mode rewriting $emr(\phi)$ of a linear equation system is an $h_\mathbb{B}$-exact formula that is $\mathbb{R}_+$-equivalent to $\phi$. We now introduce the boolean rewriting by lifting the elementary mode rewriting to a richer class of formulas. Given a vector $\mathbf{z} \in \mathcal{V}^*$, a linear equation system $\phi \in \mathcal{F}_{\Sigma_{bool}}$, and a formula $\phi' \in \mathcal{F}_{\Sigma_{bool}}$, the boolean rewriting is defined by:

$$
br(\exists \mathbf{z}.\ (\phi \wedge \phi')) =_{\text{def}} \exists \mathbf{z}.\ (emr(\phi) \wedge \phi')
$$

The boolean rewriting may indeed reduce the overapproximation coming with abstract interpretation of formulas over the booleans, as show by the following proposition.

**Proposition 8.** $h_\mathbb{B} \circ sol^{\mathbb{R}_+}(\psi) \subseteq sol^{\mathbb{B}}(br(\psi)) \subseteq sol^{\mathbb{B}}(\psi)$.

**Proof.** Let $\phi$ be a linear equation system, $\mathbf{z} \in \mathcal{V}^*$, $\phi' \in \mathcal{F}_{\Sigma_{bool}}$ and $\psi =_{\text{def}} \exists \mathbf{z}.\phi \wedge \phi'$. Since $\phi$ is $\mathbb{R}_+$-equivalent to $emr(\phi)$, it follows that $br(\psi)$ is $\mathbb{R}_+$-equivalent to $\psi$. Hence, $sol^{\mathbb{R}_+}(\psi) = sol^{\mathbb{R}_+}(br(\psi))$ so that:

$$
h_\mathbb{B} \circ sol^{\mathbb{R}_+}(\psi) = h_\mathbb{B} \circ sol^{\mathbb{R}_+}(br(\psi))
$$

By John's theorem, we have:

$$
h_\mathbb{B} \circ sol^{\mathbb{R}_+}(br(\psi)) \subseteq sol^{\mathbb{B}}(br(\psi))
$$

Furthermore, by $h_\mathbb{B}$-exactness, $\mathbb{R}_+$-equivalence, and again John's theorem, we have:

$$sol^\mathbb{B}(emr(\phi)) = h_\mathbb{B} \circ sol^{\mathbb{R}_+}(emr(\phi)) = h_\mathbb{B} \circ sol^{\mathbb{R}_+}(\phi) \subseteq sol^\mathbb{B}(\phi)$$

Therefore, it follows that:

$$sol^\mathbb{B}(br(\psi)) \subseteq sol^\mathbb{B}(\psi)$$

In combination this yields the inclusions of the proposition. □

**Theorem 5** (Main). *For any $h_\mathbb{B}$-mixed system $\psi \in \mathcal{F}_\Sigma$ the boolean rewriting $br(\psi)$ is $h_\mathbb{B}$-exact, $\mathbb{R}_+$-equivalent to $\psi$, and can be computed in at most exponential time.*

**Proof.** Let $\psi$ be a $h_\mathbb{B}$-mixed system $\exists \mathbf{x}.\ (\phi \wedge \phi')$. where $\phi$ is a linear equation system and $\phi'$ a first-order formula that is $h_\mathbb{B}$-exact and $h_\mathbb{B}$-invariant. Based on the elementary modes rewriting Corollary 1, the linear equation system $\phi$ can be transformed in at most exponential time to the form $emr(\psi) = \exists \mathbf{z}.\phi''$ where $\phi''$ is a quasipositive strongly-triangular system of linear equations. Such polynomial equation systems are $h_\mathbb{B}$-exact by Theorem 3, and so is $\phi''$. The Invariance Proposition 5 shows that the conjunction $\phi'' \wedge \phi'$ is $h_\mathbb{B}$-exact too, since $\phi'$ was assumed to be $h_\mathbb{B}$-exact and $h_\mathbb{B}$-invariant. The $h_\mathbb{B}$-exactness is preserved by existential quantification by Proposition 2, so the formula $br(\psi) = \exists \mathbf{x}.\ emr(\phi) \wedge \phi'$ is $h_\mathbb{B}$-exact too. □

**Corollary 3.** *The $h_\mathbb{B}$-abstraction of the $\mathbb{R}_+$-solution set of a $h_\mathbb{B}$-mixed system $\phi$, that is $h_\mathbb{B} \circ sol^{\mathbb{R}_+}(\phi)$, can be computed in at most exponential time in the size of the system $\phi$.*

**Proof.** Given a $h_\mathbb{B}$-mixed system $\phi$, we can apply Theorem 5 to compute in at most exponential time a $\mathbb{R}_+$-equivalent formula $\phi''$ that is $h_\mathbb{B}$-exact. It is then sufficient to compute $sol^\mathbb{B}(\phi'')$ in exponential time in the size of $\phi$. This can be done in the naive manner, that is by evaluating the formula $\phi''$—which may be of exponential size—over all possible boolean variable assignments, of which there may be exponentially many. For each assignment, the evaluation can be done in PSPACE, and thus in exponential time. The overall time required is thus a product of two exponentials, which remains exponential. □

The algorithm from the proof Corollary 3 can be improved so that it becomes sufficiently efficient for practical use. For this the two steps with exponential worst case complexity must be made polynomial for the particular instances. Firstly, note that the computation of the elementary modes (Corollary 1) is known to be computationally feasible. Various algorithms for this purpose were implemented [16,24–26] and applied successfully to problems in systems biology [14]. The second exponential step concerns the enumeration of all boolean variable assignments. This enumeration may be avoided by using constraint programming techniques for computing the solution set $sol^\mathbb{B}(\phi'')$. For those $h_\mathbb{B}$-mixed systems for which both steps can be done in polynomial time, we can compute the boolean abstraction of the $\mathbb{R}_+$-solution set in polynomial time too. The practical feasibility of this approach was demonstrated recently at an application to knockout prediction in systems biology [6], where previously only over-approximations could be computed.

## 9. Computing Sign Abstractions

We next show how to compute the sign abstraction $h_\mathbb{S} \circ sol^\mathbb{R}(\phi)$ for systems $\phi$ of linear $\Sigma_{bool}$-equations. To apply $h_\mathbb{B}$-exact rewriting, we decompose the sign abstraction into the boolean abstraction and functions definable in first-order logic.

### 9.1. Decomposition

We can decompose any real number $r \in \mathbb{R}$ into a pair of two positive numbers $dec(r) \in \mathbb{R}_+^2$—negative and the positive part—as follows:

$$dec(r) =_{\text{def}} \begin{cases} (0, r) & \text{if } r \geq 0 \\ (-r, 0) & \text{if } r \leq 0 \end{cases}$$

The image of this surjective function is $\{0\} \times \mathbb{R}_+) \cup (\mathbb{R}_+ \times \{0\}$, so it has an inverse $\text{dec}^{-1} : (\{0\} \times \mathbb{R}_+) \cup (\mathbb{R}_+ \times \{0\}) \to \mathbb{R}$, which satisfies for all pairs $(r_1, r_2)$ in the domain:

$$\text{dec}^{-1}(r_1, r_2) = r_2 -^{\mathbb{R}} r_1$$

Furthermore, recall that $h_{\mathbb{B}}^2 : \mathbb{R}_+^2 \to \mathbb{B}^2$ satisfies $h_{\mathbb{B}}^2(r_1, r_2) = (h_{\mathbb{B}}(r_1), h_{\mathbb{B}}(r_2))$.

**Lemma 18** (Decomposition). $h_{\mathbb{S}} = dec^{-1} \circ h_{\mathbb{B}}^2 \circ dec$

**Proof.** If $r$ is negative then $\text{dec}^{-1}(h_{\mathbb{B}}^2(\text{dec}(r))) = \text{dec}^{-1}(h_{\mathbb{B}}^2((-r, 0))) = \text{dec}^{-1}((h_{\mathbb{B}}(-r), 0))$ $= -h_{\mathbb{B}}(-r) = h_{\mathbb{S}}(r)$. Otherwise if $r$ is positive then $\text{dec}^{-1}(h_{\mathbb{B}}^2(\text{dec}(r))) = \text{dec}^{-1}(h_{\mathbb{B}}^2((0, r)))$ $= \text{dec}^{-1}((0, h_{\mathbb{B}}(r)) = h_{\mathbb{B}}(r) = h_{\mathbb{S}}(r)$. $\square$

### 9.2. Positivity

We show in a first step that first-order formulas over the reals can be rewritten, such that interpretation over the positive reals is enough.

We call a formula $\phi \in \mathcal{F}_{\Sigma_{bool}}$ flat if all equations contained in $\phi$ have the form $x \overset{\circ}{=} x_1 + x_2$, $x \overset{\circ}{=} x_1 * x_2$, $x \overset{\circ}{=} 0$, or $x \overset{\circ}{=} 1$ for some variables $x, x_1, x_2$. Note that any formula $\phi \in \mathcal{F}_{\Sigma_{bool}}$ can be converted to an equivalent flat formula in linear time by introducing fresh existentially quantified variables, so that we can assume flatness without loss of generality.

We fix two generators of fresh variable $\nu_{\ominus}, \nu_{\oplus} : \mathcal{V} \to \mathcal{V}$. For any $x \in \mathcal{V}$, the intention is that $\nu_{\oplus}(x)$ stands for the positive part of $x$ and $\nu_{\ominus}(x)$ for its negative part. We will preserve the invariants $x = \nu_{\oplus}(x) - \nu_{\ominus}(x)$ and $\nu_{\oplus}(x) * \nu_{\ominus}(x) = 0$. Furthermore, we define $\nu : \mathcal{V} \to \mathcal{V}^2$ such that for all $x \in \mathcal{V}$:

$$\nu(x) =_{\text{def}} (\nu_{\ominus}(x), \nu_{\oplus}(x))$$

For any flat formula $\phi \in \mathcal{F}_{\Sigma}(V)$ we define a formula $\text{dec}_{\nu}(\phi) \in \mathcal{F}_{\Sigma}(\nu_{\ominus}(V) \cup \nu_{\oplus}(V))$ with the variables $\nu_{\ominus}(x)$ and $\nu_{\oplus}(x)$ instead of $x$ for all $x \in V$. Otherwise the formula $\widetilde{\text{dec}}_{\nu}(\phi)$ has the same meaning as over the reals than $\phi$.

$$\widetilde{\text{dec}}_{\nu}(\phi) = \text{dec}_{\nu}(\phi) \wedge \bigwedge_{x \in V} \nu_{\oplus}(x) * \nu_{\ominus}(x) \overset{\circ}{=} 0$$

where

$\text{dec}_{\nu}(x \overset{\circ}{=} x_1 + x_2) =$
  $\quad \nu_{\oplus}(x) + \nu_{\ominus}(x_1) + \nu_{\ominus}(x_2) \overset{\circ}{=}$
  $\quad \nu_{\ominus}(x) + \nu_{\oplus}(x_1) + \nu_{\oplus}(x_2)$
$\text{dec}_{\nu}(x \overset{\circ}{=} 0) = \nu_{\oplus}(x) \overset{\circ}{=} \nu_{\ominus}(x)$
$\text{dec}_{\nu}(\exists x.\phi) = \exists \nu_{\ominus}(x).\exists \nu_{\oplus}(x).$
  $\quad \nu_{\oplus}(x) * \nu_{\ominus}(x) \overset{\circ}{=} 0 \wedge \text{dec}_{\nu}(\phi)$

$\text{dec}_{\nu}(x \overset{\circ}{=} x_1 * x_2) =$
  $\quad \nu_{\oplus}(x) + \nu_{\oplus}(x_1) * \nu_{\ominus}(x_2) + \nu_{\ominus}(x_1) * \nu_{\oplus}(x_2) \overset{\circ}{=}$
  $\quad \nu_{\ominus}(x) + \nu_{\oplus}(x_1) * \nu_{\oplus}(x_2) + \nu_{\ominus}(x_1) * \nu_{\ominus}(x_2)$
$\text{dec}_{\nu}(x \overset{\circ}{=} 1) = \nu_{\oplus}(x) \overset{\circ}{=} \nu_{\ominus}(x) + 1$
$\text{dec}_{\nu}(\phi \wedge \phi') = \text{dec}_{\nu}(\phi) \wedge \text{dec}_{\nu}(\phi')$
$\text{dec}_{\nu}(\neg \phi) = \neg \text{dec}_{\nu}(\phi)$

Note that the definition in the case of addition, the definition relies on that subtraction $-^{\mathbb{R}}$ in the structure of reals is the inverse of addition $+^{\mathbb{R}}$. The expressions that are to be subtracted on one side of the equation are added to the other side instead. This is also used in the case of multiplication, in combination with the distributivity law for addition $+^{\mathbb{R}}$

and multiplication $*^{\mathbb{R}}$. Furthermore, $\widetilde{dec}_\nu(\phi)$ belongs to $\mathcal{F}_{\Sigma_{bool}}(\nu_\ominus(V) \cup \nu_\ominus(V))$ and can be computed in linear time from $\phi$.

**Proposition 9** (Positivity). *For any flat formula $\phi \in \mathcal{F}_{\Sigma_{bool}}(V)$:*

$$dec \circ sol_V^{\mathbb{R}}(\phi) = \{\sigma^2 \circ \nu_{|V} \mid \sigma \in sol^{\mathbb{R}+}(\widetilde{dec}_\nu(\phi))\}$$

**Proof.** By induction on the structure of $\phi$. In the first case of reals, can use that $-^{\mathbb{R}}$ is the inverse of $+^{\mathbb{R}}$ and that the distributivity laws holds for $+^{\mathbb{R}}$ and $*^{\mathbb{R}}$. $\square$

**Lemma 19.** *For any flat linear equation system $\phi$, the formula $\widetilde{dec}_\nu(\phi)$ is a $h_{\mathbb{B}}$-mixed system.*

**Proof.** If $\phi$ is a flat linear system, then $dec_\nu(\phi)$ is a linear system, so that $\widetilde{dec}_\nu(\phi)$ is a $h_{\mathbb{B}}$-mixed system. $\square$

*9.3. Computing Sign Abstractions*

We now have developed all the prerequisite for computing the sign abstraction of linear equation systems by using $h_{\mathbb{B}}$-exact boolean rewriting of $h_{\mathbb{B}}$-mixed systems.

**Theorem 6.** *For any linear equation system $\phi \in \mathcal{F}_{\Sigma_{bool}}(V)$, the formula $br(\widetilde{dec}_\nu(\phi))$ can be computed in at most exponential time and satisfies:*

$$h_{\mathbb{S}} \circ sol_V^{\mathbb{R}}(\phi) = \{[y / \tau(\nu_\oplus(y)) -^{\mathbb{R}} \tau(\nu_\ominus(y)) \mid y \in V] \mid \tau \in sol^{\mathbb{B}}(br(\widetilde{dec}_\nu(\phi)))\}$$

**Proof.** Let $\phi \in \mathcal{F}_{\Sigma_{bool}}(V)$ be a system of linear equations. Without loss of generality, we can assume that $\phi$ is flat. Let: $\tilde{\phi} =_{\text{def}} \widetilde{dec}_\nu(\phi)$. The formula $\tilde{\phi}$ is a $h_{\mathbb{B}}$-mixed system by Lemma 19 with $fv(\tilde{\phi}) = \nu_\ominus(V) \cup \nu_\oplus(V)$ so that we can apply the Main Theorem 5 to it. It shows that boolean rewriting $br(\tilde{\phi})$ is an $\mathbb{R}_+$-equivalent formula in $\mathcal{F}_\Sigma(\nu_\oplus(V) \cup \nu_\ominus(V))$ that is $h_{\mathbb{B}}$-exact and can be computed in at most exponential time. We can now conclude as follows:

$$
\begin{aligned}
&h_{\mathbb{S}} \circ sol_V^{\mathbb{R}}(\phi) \\
&= \quad dec^{-1} \circ h_{\mathbb{B}}^2 \circ dec \circ sol_V^{\mathbb{R}}(\phi) && \text{Decomposition Lemma 18} \\
&= \quad dec^{-1} \circ h_{\mathbb{B}}^2 \circ \{\sigma^2 \circ \nu_{|V} \mid \sigma \in sol^{\mathbb{R}+}(\tilde{\phi})\} && \text{Positivity Proposition 9} \\
&= \quad dec^{-1} \circ h_{\mathbb{B}}^2 \circ \{\sigma^2 \circ \nu_{|V} \mid \sigma \in sol^{\mathbb{R}+}(br(\tilde{\phi}))\} && \mathbb{R}_+\text{-equivalence of } \tilde{\phi} \text{ and } br(\tilde{\phi}) \\
&= \quad \{dec^{-1} \circ \tau^2 \circ \nu_{|V} \mid \tau \in sol^{\mathbb{B}}(br(\tilde{\phi}))\} && h_{\mathbb{B}}\text{-exactness of } br(\tilde{\phi}) \\
&= \quad \{[y / \tau(\nu_\oplus(y)) -^{\mathbb{R}} \tau(\nu_\ominus(y)) \mid y \in V] && \text{definition of } dec^{-1} \\
&\qquad\qquad \mid \tau \in sol^{\mathbb{B}}(br(\tilde{\phi}))\}
\end{aligned}
$$

$\square$

The sign abstraction of a system $\phi$ of $\Sigma_{bool}$-equations with free variables in $V = fv(\phi)$ can thus be computed by first computing the $h_{\mathbb{B}}$-exact formula $br(\tilde{\phi}) \in \mathcal{F}_\Sigma(\nu_\oplus(V) \cup \nu_\ominus(V))$ from Theorem 6 by applying the Positivity Proposition 9 and the Main Theorem 5, then computing $sol^{\mathbb{B}}(br(\tilde{\phi}))$ by finite domain constraint programming, and finally inferring $h_{\mathbb{S}} \circ sol^{\mathbb{R}}(\phi)$ thereof based on the equation of Theorem 6.

**Corollary 4.** *The sign abstraction $h_{\mathbb{S}} \circ sol_V^{\mathbb{R}}(\phi)$ can be computed in at most single exponential time in the size of $\phi$.*

**Proof.** The formula $br(\tilde{\phi})$ is of exponential size but contains only twice as many variables than $\phi$. Let $n = |fv(\phi)|$. We can then compute $h_{\mathbb{S}} \circ sol_V^{\mathbb{R}}(\phi)$ by testing $6^{2n}$ variable assignments for membership to $sol^{\mathbb{R}}(br(\tilde{\phi}))$. Each such test is linear in the size of $br(\tilde{\phi})$, and thus

in $O(2^m)$ where $m$ is the size of $\phi$. So the overall time is in $O(6^{2n}2^m)$ and since $n \le m$ in $O(6^{3m})$. $\quad\square$

We finally show that the same algorithm as for computing the sign abstraction for linear equation systems can be lifted to a richer class of formulas to obtain another and possibly more precise overapproximation of the sign abstraction than John's.

**Proposition 10.** *Let $\psi = \exists z.\ \phi \wedge \phi'$ in $\mathcal{F}_{\Sigma_{bool}}(V)$ for some linear equation system $\phi$ and formula $\phi' \in \widetilde{\mathcal{F}_{\Sigma_{bool}}}$. The formula $br(\widetilde{dec}_\nu(\psi))$ then yields an overapproximation of the sign abstraction of $\phi$:*

$$h_{\mathbb{S}} \circ sol_V^{\mathbb{R}}(\psi) \subseteq \{[y/\tau(\nu_\oplus(y)) -^{\mathbb{R}} \tau(\nu_\ominus(y)) \mid y \in V] \mid \tau \in sol^{\mathbb{B}}(br(\widetilde{dec}_\nu(\psi)))\}$$

**Proof.** Along the lines of the proof of Theorem except that $br(\widetilde{dec}_\nu(\psi))$ is not $h_{\mathbb{B}}$-exact. Therefore, the equality where the $h_{\mathbb{B}}$-exactness was used must be weakened to an inclusion. $\quad\square$

## 10. Application to Program Analysis

We illustrate our results by applying the sign abstraction for program analysis based on abstract interpretation. We consider the Python implementation in Figure 6 of the function $\text{I} : \mathbb{R}^2 \to \mathbb{R}$. A call $\text{I}(a, s)$ supposedly computes the approximation of the integral $\int_0^a \text{f}(x)dx$ with step width $s$ for some total function $\text{f} : \mathbb{R} \to \mathbb{R}$. Abstract interpretation allows us to find out the conditions that must hold on the input parameters for $\text{I}((a : float, s : float)$ to work properly, and in particular to avoid exception throwing.

```python
def I(a: float , s: float ):
    if a < 0: raise ValueError ('This should never happen ')
    if s > a:
        return 0
    else :
        return s * f(a) + I(a - s, s)
```

**Figure 6.** Python function approximating the integral $\int_0^a \text{f}(x)dx$ for a given function $\text{f} : \mathbb{R} \to \mathbb{R}$.

We can first interpret numeric programs abstractly as a formula of first-order logic with signature $\Sigma_{arith}$. We illustrate this in an ad hoc manner on the integral example $\text{I}$:

$$\phi_{\text{I}} =_{\text{def}} \begin{aligned} &\exists ret_{\text{f}} \exists ret_{\text{I}} \exists result. \\ &(a < 0 \iff raise\_exception \overset{\circ}{=} 1) \wedge \\ &((s > a \wedge do\_recursion \overset{\circ}{=} 0 \wedge result \overset{\circ}{=} 0) \vee \\ &(\neg(s > a) \wedge do\_recursion \overset{\circ}{=} 1 \wedge a_{rec} \overset{\circ}{=} a - s \wedge s_{rec} \overset{\circ}{=} s \wedge \\ &\quad result \overset{\circ}{=} s \cdot ret_{\text{f}} + ret_{\text{I}})) \end{aligned}$$

The variables $a$ and $s$ are the formal parameters in the definition of $\text{I}(a : float, s : float)$. The others are fresh variables introduced to handle exceptions or function calls: the boolean flag $raise\_exception$ represents exception throwing, the boolean flag $do\_recursion$ has a true value only when a recursive call is made to $\text{I}$ with actual parameters represented by the variables $a_{rec}, s_{rec}$ and return value represented by $ret_{\text{I}}$, while $ret_{\text{f}}$ is the variable for the return value of the call to the function $\text{f}$. The final return value of $\text{I}$ is represented by the variable $result$. In what follows, we are not interested in the signs of the last three variables, so we quantify them existentially.

The sign behavior of function $\text{I}$ is given by the formula's sign abstraction $h_{\mathbb{S}} \circ sol^{\mathbb{R}}(\phi_{\text{I}})$. Given that $\phi_{\text{I}}$ is not $h_{\mathbb{B}}$-mixed system, we cannot apply the algorithm from Theorem 6 directly to compute this sign abstraction. Nevertheless, it will be beneficial as we will illustrate below.

By John's theorem, the sign abstraction $h_{\mathbb{S}} \circ sol^{\mathbb{R}}(\phi_{\mathtt{I}})$ can be overapproximated by the abstract interpretation $sol^{\mathbb{S}}(\phi_{\mathtt{I}})$. Since $\mathbb{S}$ is a finite structure, this abstract interpretation can be computed by finite domain constraint programming. For this, we implemented a solver for first-order formulas over the structure $\mathbb{S}$ with Minizinc [17]. When applied to $\phi_{\mathtt{I}}$ it returns the set of abstract solutions $sol^{\mathbb{S}}(\phi_{\mathtt{I}})$ given in Table 1. This set contains the 6 unjustified abstract solutions $2, 4, 10, 13, 15, 18$ outside $h_{\mathbb{S}} \circ sol^{\mathbb{R}}(\phi_{\mathtt{I}})$. In the table they are distinguished by gray background color. We also note that the last three solutions $17, 18, 19$ could be ruled out when using a more precise abstract program interpretation, taking into account that no recursive call is possible when an exception is thrown.

**Table 1.** Set of abstract solutions in $sol^{\mathbb{S}}(\phi_{\mathtt{I}})$. Six solutions with gray background color are unjustified since outside $h_{\mathbb{S}} \circ sol^{\mathbb{R}}(\phi_{\mathtt{I}})$.

| # | *raise_exception* | *do_recursion* | *a* | *s* | $a_{rec}$ | $s_{rec}$ |
|---|---|---|---|---|---|---|
| 1. | 0 | 0 | 0 | 1 | −1 | 1 |
| 2. | 0 | 0 | 1 | 1 | 0 | 1 |
| 3. | 0 | 0 | 1 | 1 | −1 | 1 |
| 4. | 0 | 0 | 1 | 1 | 1 | 1 |
| 5. | 0 | 1 | 0 | 0 | 0 | 0 |
| 6. | 0 | 1 | 1 | 0 | 1 | 0 |
| 7. | 0 | 1 | 0 | −1 | 1 | −1 |
| 8. | 0 | 1 | 1 | 1 | 0 | 1 |
| 9. | 0 | 1 | 1 | −1 | 1 | −1 |
| 10. | 0 | 1 | 1 | 1 | −1 | 1 |
| 11. | 0 | 1 | 1 | 1 | 1 | 1 |
| 12. | 1 | 0 | −1 | 0 | −1 | 0 |
| 13. | 1 | 0 | −1 | −1 | 0 | −1 |
| 14. | 1 | 0 | −1 | −1 | −1 | −1 |
| 15. | 1 | 0 | −1 | −1 | 1 | −1 |
| 16. | 1 | 0 | −1 | 1 | −1 | 1 |
| 17. | 1 | 1 | −1 | −1 | 0 | −1 |
| 18. | 1 | 1 | −1 | −1 | −1 | −1 |
| 19. | 1 | 1 | −1 | −1 | 1 | −1 |

The sets of abstract solutions provide information on possible sign of values of the parameters in a call $\mathtt{I}(a : float, s : float)$. For example, solution 1 in Table 1 states that when called with values of signs $[a/0, s/1]$ the function $\mathtt{I}$ will not raise an exceptions nor make a recursive call. Solution 8 states that when called with values of signs $[a/1, s/1]$ function $\mathtt{I}$ may go into recursion with signs $[a_{rec}/0, s_{rec}/1]$ without raising an exception.

Any set of abstract solutions defines an abstract call graph. The abstract call graphs of $sol^{\mathbb{S}}(\phi_{\mathtt{I}})$ and $h_{\mathbb{S}} \circ sol^{\mathbb{R}}(\phi_{\mathtt{I}})$ from Table 1 are given in Figure 7. Solution 1 in Table 1 implies a solid edge from the node $\mathtt{I}^{\mathbb{S}}(1,1)$ to the node $\mathtt{I}^{\mathbb{S}}(0,1)$. The edge is solid since solution 1 is justified. Edges induced by unjustified solutions are dashed. The unjustified solution 10 for instance induces the dashed edge from $\mathtt{I}^{\mathbb{S}}(1,1)$ to $\mathtt{I}^{\mathbb{S}}(1,-1)$. Solutions with *do_recursion* $= 0$ and *raise_exception* $= 0$ do not induce any edge. Instead, they show that the computation may stop, producing final nodes that are surrounded by a double circle. The final nodes are $\mathtt{I}^{\mathbb{S}}(1,1)$ and $\mathtt{I}^{\mathbb{S}}(0,1)$. Note that for all nonfinal nodes, either an exception is raised or the computation loops endlessly. Solutions with *raise_exception* $= 1$ induce an edge to the EXCEPT node.

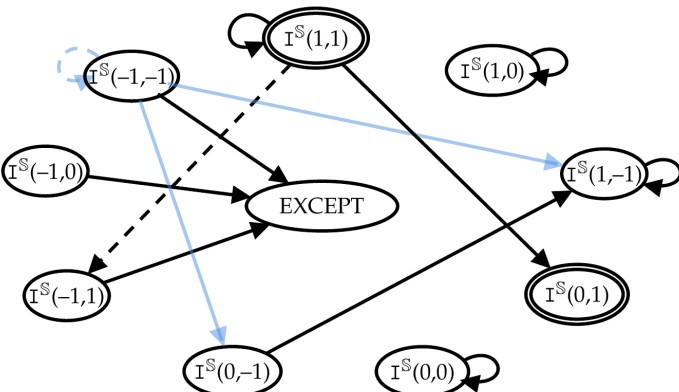

**Figure 7.** Sign call graph of function I in Figure 6 created from the sets of abstract solutions in Table 1. Solid lines correspond to abstract solutions in $h_{\mathbb{S}} \circ sol^{\mathbb{R}}(\phi_{\mathtt{I}})$, while dashed lines correspond to unjustified abstract solutions in $sol^{\mathbb{S}}(\phi_{\mathtt{I}})$. For example, $\mathtt{I}^{\mathbb{S}}(1, -1)$ represents assignment $[a/1, s/-1]$, that is signs of $a$ and $s$ in calls $\mathtt{I}(a : float, s : float)$ where $a > 0$ and $s < 0$. Light blue edges may be removed by improving $\phi_{\mathtt{I}}$ so that solutions 17, 18, 19 become impossible. Computation may terminate without raising an exception in nodes surrounded by a double circle.

Given that only 2 unjustified solutions with *do_recursion* $= 0$ and *raise_exception* $= 0$ (10 and 18), there are only 2 dashed edges in the graph. Furthermore, the edges induced by the last three solutions 17, 18, 19 are drawn in blue, since these could be removed with a more precise abstract program interpretation than $\phi_{\mathtt{I}}$.

The sign analysis without the unjustified dashed edges yields the following result: the program in state $\mathtt{I}^{\mathbb{S}}(1, 1)$, where $a > 0$ and $s > 0$ may either terminate, loop indefinitely, or go to state $\mathtt{I}^{\mathbb{S}}(0, 1)$ and terminate there immediately. With the unjustified dashed edges, however, it wrongly seems possible that the program may also raise an exception by passing through $\mathtt{I}^{\mathbb{S}}(-1, 1)$. This overapproximation would be particularly unfortunate since state $\mathtt{I}^{\mathbb{S}}(1, 1)$ is the only useful state to call I.

We next show how to remove the unjustified solutions by applying the overapproximation algorithm for the sign abstraction from Proposition 10, that lifts the algorithm for exact sign abstraction from Theorem 6 to a richer class of formulas. The idea is to split the formula $\phi_{\mathtt{I}}$ into its linear part and the rest. Before doing so, we preprocess the inequation $s > a$: We introduce a fresh variable *signvar*, add the equation $s - a \stackrel{\circ}{=} signvar$, and rewrite $s > a$ to *signvar* $> 0$. The linear part of $\phi_{\mathtt{I}}$ then becomes:

$$s - a \stackrel{\circ}{=} signvar \wedge a_{rec} \stackrel{\circ}{=} a - s \wedge s_{rec} \stackrel{\circ}{=} s$$

We can then rewrite the linear part into the signature $\Sigma_{bool}$ by moving the negative parts positively onto the other side. This yields the following linear equation system:

$$s \stackrel{\circ}{=} signvar + a \wedge a_{rec} + s \stackrel{\circ}{=} a \wedge s_{rec} \stackrel{\circ}{=} s$$

The remainder of $\phi_{\mathtt{I}}$ can be rewritten as follows:

$$((a < 0 \wedge raise\_exception > 0) \vee (a \geq 0 \wedge raise\_exception \stackrel{\circ}{=} 0))$$
$$\wedge ((signvar > 0 \wedge do\_recursion \stackrel{\circ}{=} 0 \wedge result \stackrel{\circ}{=} 0) \vee$$
$$(signvar \leq 0 \wedge do\_recursion > 0 \wedge result \stackrel{\circ}{=} s * ret_{\mathtt{f}} + ret_{\mathtt{I}}))$$

It is not clear whether the conjunction of both parts is a $h_{\mathbb{B}}$-mixed system, since it is not clear how to show the $h_{\mathbb{B}}$-invariance of the equation $result \stackrel{\circ}{=} s * ret_{\mathtt{f}} + ret_{\mathtt{I}}$. Still, we can apply the overapproximation algorithm of the sign abstraction from Proposition 10. It indeed improves on John's approximation, ruling out both unjustified solutions. The details are worked out in Appendix A.

In the general case, linear equation systems are not enough, in which case our algorithm from Theorem 6 for computing sign abstractions cannot be applied. But then we can still apply the overapproximation algorithm from Proposition 10 which rewrites a linear part of the formula exactly. As illustrated by the present example, this overapproximation is often way more precise than John's.

## 11. Example for the Overapproximation of the Sign Abstraction

We reconsider conjunction of the linear part obtained and the rest of $\phi_{\mathtt{I}}$, that is $\phi_{\mathtt{I}}^{lin} \wedge \phi_{\mathtt{I}}^{rest}$ where:

$$
\phi_{\mathtt{I}}^{lin} =_{\mathrm{def}} \left\{
\begin{array}{rl}
& s \overset{\circ}{=} signvar + a \\
\wedge & a_{rec} + s \overset{\circ}{=} a \\
\wedge & s_{rec} \overset{\circ}{=} s
\end{array}
\right.
$$

$$
\phi_{\mathtt{I}}^{rest} =_{\mathrm{def}} \left\{
\begin{array}{rl}
( & (a < 0 \wedge raise\_exception > 0) \\
\vee & (a \geq 0 \wedge raise\_exception \overset{\circ}{=} 0)) \\
\wedge \quad ( & (signvar > 0 \wedge do\_recursion \overset{\circ}{=} 0 \wedge result \overset{\circ}{=} 0) \\
\vee & (signvar \leq 0 \wedge do\_recursion > 0 \wedge result \overset{\circ}{=} s * ret_{\mathtt{f}} + ret_{\mathtt{I}}))
\end{array}
\right.
$$

The decomposition of the linear subsystem $\mathrm{dec}_\nu(\phi_{\mathtt{I}}^{lin})$ for interpretation over $\mathbb{B}$ as defined in Section 9 is obtained by splitting each variable $x$ into two fresh variables $\nu_\oplus(x)$ and $\nu_\ominus(x)$ representing its positive and negative part:

$$
\mathrm{dec}_\nu(\phi_{\mathtt{I}}^{lin}) = \left\{
\begin{array}{rl}
& \nu_\oplus(s) + \nu_\ominus(a) + \nu_\ominus(signvar) \overset{\circ}{=} \nu_\ominus(s) + \nu_\oplus(a) + \nu_\oplus(signvar) \\
\wedge & \nu_\oplus(a_{rec}) + \nu_\ominus(a) + \nu_\oplus(s) \overset{\circ}{=} \nu_\ominus(a_{rec}) + \nu_\oplus(a) + \nu_\ominus(s) \\
\wedge & \nu_\oplus(s_{rec}) + \nu_\ominus(s) \overset{\circ}{=} \nu_\ominus(s_{rec}) + \nu_\oplus(s)
\end{array}
\right.
$$

The additional constraints on the decomposition variables are:

$$
\begin{array}{rl}
& \nu_\oplus(s) * \nu_\ominus(s) \overset{\circ}{=} 0 \\
\wedge & \nu_\oplus(a) * \nu_\ominus(a) \overset{\circ}{=} 0 \\
\wedge & \nu_\oplus(signvar) * \nu_\ominus(signvar) \overset{\circ}{=} 0 \\
\wedge & \nu_\oplus(a_{rec}) * \nu_\ominus(a_{rec}) \overset{\circ}{=} 0 \\
\wedge & \nu_\oplus(s_{rec}) * \nu_\ominus(s_{rec}) \overset{\circ}{=} 0 \\
\wedge & \nu_\oplus(result) * \nu_\ominus(result) \overset{\circ}{=} 0 \\
\wedge & \nu_\oplus(ret_{\mathtt{I}}) * \nu_\ominus(ret_{\mathtt{I}}) \overset{\circ}{=} 0 \\
\wedge & \nu_\oplus(ret_{\mathtt{f}}) * \nu_\ominus(ret_{\mathtt{f}}) \overset{\circ}{=} 0
\end{array}
$$

The elementary mode rewriting $emr(\mathrm{dec}_\nu(\phi_{\mathtt{I}}^{lin}))$ is the following $\mathbb{R}_+$-equivalent $h_{\mathbb{B}}$-exact $\Sigma_{bool}$-formula obtained via Corollary 1:

$$
\begin{array}{rl}
\exists x_0 \ldots \exists x_{10}. & \\
\wedge & \nu_\ominus(a) \overset{\circ}{=} x_{10} + x_8 + x_9 \\
\wedge & \nu_\oplus(a) \overset{\circ}{=} x_{10} + x_6 + x_7 \\
\wedge & \nu_\ominus(a_{rec}) \overset{\circ}{=} x_4 + x_5 + x_9 \\
\wedge & \nu_\oplus(a_{rec}) \overset{\circ}{=} x_3 + x_5 + x_7 \\
\wedge & \nu_\ominus(signvar) \overset{\circ}{=} x_2 + x_3 + x_7 \\
\wedge & \nu_\oplus(signvar) \overset{\circ}{=} x_2 + x_4 + x_9 \\
\wedge & \nu_\ominus(s) \overset{\circ}{=} x_1 + x_3 + x_8 \\
\wedge & \nu_\oplus(s) \overset{\circ}{=} x_1 + x_4 + x_6 \\
\wedge & \nu_\ominus(s_{rec}) \overset{\circ}{=} x_0 + x_3 + x_8 \\
\wedge & \nu_\oplus(s_{rec}) \overset{\circ}{=} x_0 + x_4 + x_6
\end{array}
$$

The nonlinear remainder also needs to be rewritten with the decomposition variables for interpretation over $\mathbb{B}$. The formula below is $\mathrm{dec}_\nu(\phi_{\mathtt{I}}^{lin})$ except that we simplified the rewriting of inequations a bit.

$$
\begin{aligned}
( \quad & (\neg \nu_\ominus(a) \overset{\circ}{=} 0 \wedge \neg \nu_\oplus(\mathit{raise\_exception}) \overset{\circ}{=} 0) \\
\vee \quad & (\nu_\ominus(a) \overset{\circ}{=} 0 \wedge \nu_\ominus(\mathit{raise\_exception}) \overset{\circ}{=} 0 \wedge \nu_\oplus(\mathit{raise\_exception}) \overset{\circ}{=} 0)) \\
\wedge \quad ( \quad & (\neg \nu_\oplus(\mathit{signvar}) \overset{\circ}{=} 0 \wedge \nu_\ominus(\mathit{do\_recursion}) \overset{\circ}{=} 0 \wedge \nu_\oplus(\mathit{do\_recursion}) \overset{\circ}{=} 0 \\
& \wedge \nu_\ominus(\mathit{result}) \overset{\circ}{=} 0 \wedge \nu_\oplus(\mathit{result}) \overset{\circ}{=} 0) \\
\vee \quad & (\nu_\oplus(\mathit{signvar}) \overset{\circ}{=} 0 \wedge \neg \nu_\oplus(\mathit{do\_recursion}) \overset{\circ}{=} 0 \\
& \wedge \quad \nu_\oplus(\mathit{result}) + \nu_\ominus(s) * \nu_\oplus(\mathit{ret}_{\mathtt{f}}) + \nu_\oplus(s) * \nu_\ominus(\mathit{ret}_{\mathtt{f}}) + \nu_\ominus(\mathit{ret}_{\mathtt{I}})) \\
& \overset{\circ}{=} \nu_\ominus(\mathit{result}) + \nu_\oplus(s) * \nu_\oplus(\mathit{ret}_{\mathtt{f}}) + \nu_\ominus(s) * \nu_\ominus(\mathit{ret}_{\mathtt{f}}) + \nu_\oplus(\mathit{ret}_{\mathtt{I}})))
\end{aligned}
$$

For any solution $\tau$ of the conjunction of the above three blocks of formulas over the algebra of booleans $\mathbb{B}$, we then obtain an assignment $\sigma \in h_{\mathbb{S}} \circ sol^{\mathbb{R}}(\phi_{\mathtt{I}})$ according to Theorem 6:

$$
\begin{aligned}
\sigma(s) &= \tau(\nu_\oplus(s)) -^{\mathbb{R}} \tau(\nu_\ominus(s)) \\
\sigma(a) &= \tau(\nu_\oplus(a)) -^{\mathbb{R}} \tau(\nu_\ominus(a)) \\
\sigma(\mathit{signvar}) &= \tau(\nu_\oplus(\mathit{signvar})) -^{\mathbb{R}} \tau(\nu_\ominus(\mathit{signvar})) \\
\sigma(a_{rec}) &= \tau(\nu_\oplus(a_{rec})) -^{\mathbb{R}} \tau(\nu_\ominus(a_{rec})) \\
\sigma(s_{rec}) &= \tau(\nu_\oplus(s_{rec})) -^{\mathbb{R}} \tau(\nu_\ominus(s_{rec})) \\
\sigma(\mathit{result}) &= \tau(\nu_\oplus(\mathit{result})) -^{\mathbb{R}} \tau(\nu_\ominus(\mathit{result})) \\
\sigma(\mathit{ret}_{\mathtt{f}}) &= \tau(\nu_\oplus(\mathit{ret}_{\mathtt{f}})) -^{\mathbb{R}} \tau(\nu_\ominus(\mathit{ret}_{\mathtt{f}})) \\
\sigma(\mathit{ret}_{\mathtt{I}}) &= \tau(\nu_\oplus(\mathit{ret}_{\mathtt{I}})) -^{\mathbb{R}} \tau(\nu_\ominus(\mathit{ret}_{\mathtt{I}}))
\end{aligned}
$$

## 12. Conclusions and Future Work

We showed that any $h_{\mathbb{B}}$-mixed system can be rewritten into an $h_{\mathbb{B}}$-exact formula by computing the elementary modes of the linear subsystem. In previous work, $h_{\mathbb{B}}$-exact rewriting $h_{\mathbb{B}}$-mixed systems was applied to compute difference abstractions exactly. In the present paper, we showed that $h_{\mathbb{B}}$-exact rewriting can also be used to compute sign-abstractions exactly.

We have illustrated the usefulness of the computation of sign abstraction for linear formulas for the sign analysis of function programs. Using John's overapproximation is often not good enough for such applications, since the relationships between the signs of different variables are quickly lost. We saw that elementary mode rewriting yields better a better approximation of the sign abstraction even for nonlinear equation systems, which may preserve these relationships.

The time for computing abstractions exactly strongly depends on the time needed to compute the elementary modes. Some experiments were reported in [6] in the case of the difference abstraction. There, one has to compute the elementary modes for a linear equation system that contains two copies of the linear equation system given with the input. The copying doubles the size and may increase the time for the computation of the elementary modes seriously. In the application of difference abstraction to change prediction of reaction networks, we observed cases where John's overapproximation of the difference abstraction could be computed in circa 10 min, while the exact computation required circa 10 h.

In the future, it would we of interest to find heuristics for approximating abstractions of linear equation systems that reduce the computation time of the exact algorithm while improving John's overapproximation in precision. In the case of difference abstractions, the minimal support heuristics was proposed for this purpose [6]. In the example mentioned above, this heuristics could be computed in circa 10 min, like John's overapproximation, while yielding the exact result. In general, however, the minimal support heuristics is not exact.

Another interesting question for future work is how to compute more quantitative abstractions exactly, as for instance with intervals. In this case however the structure of

abstract values is infinite, therefore finite domain constraint programming is no longer sufficient to compute the set of abstract solutions.

**Author Contributions:** These authors contributed equally to this work. All authors have read and agreed to the published version of the manuscript.

**Funding:** This research received no external funding.

**Data Availability Statement:** Data sharing not applicable.

**Acknowledgments:** We would like to acknowledge the reviewers for the constructive feedback.

**Conflicts of Interest:** The authors declare no conflict of interest.

## Appendix A

The system of linear $\Sigma_{bool}$-equations $dec_\nu(\phi_{\mathbb{I}}^{lin})$ corresponds to the following linear integer matrix equation:

$$
\begin{pmatrix}
1 & -1 & 0 & 0 & 1 & -1 & -1 & 1 & 0 & 0 \\
-1 & 1 & 1 & -1 & 0 & 0 & 1 & -1 & 0 & 0 \\
0 & 0 & 0 & 0 & 0 & 0 & 1 & -1 & -1 & 1
\end{pmatrix}
\begin{pmatrix}
\nu_\ominus(a) \\
\nu_\oplus(a) \\
\nu_\ominus(a_{rec}) \\
\nu_\oplus(a_{rec}) \\
\nu_\ominus(signvar) \\
\nu_\oplus(signvar) \\
\nu_\ominus(s) \\
\nu_\oplus(s) \\
\nu_\ominus(s_{rec}) \\
\nu_\oplus(s_{rec})
\end{pmatrix}
\stackrel{\circ}{=}
\begin{pmatrix}
0 \\
0 \\
0 \\
0 \\
0 \\
0 \\
0 \\
0 \\
0 \\
0
\end{pmatrix}
$$

The elementary mode rewriting $emr(dec_\nu(\phi_{\mathbb{I}}^{lin}))$ corresponds to the linear integer matrix equation :

$$
\begin{pmatrix}
0 & 0 & 1 & 0 & 0 & 0 & 0 & 0 & 0 & 1 & 1 \\
0 & 0 & 1 & 0 & 0 & 0 & 0 & 1 & 1 & 0 & 0 \\
0 & 0 & 0 & 0 & 0 & 1 & 1 & 0 & 0 & 0 & 1 \\
0 & 0 & 0 & 0 & 1 & 0 & 1 & 0 & 1 & 0 & 0 \\
0 & 0 & 0 & 1 & 1 & 0 & 0 & 0 & 1 & 0 & 0 \\
0 & 0 & 0 & 1 & 0 & 1 & 0 & 0 & 0 & 0 & 1 \\
0 & 1 & 0 & 0 & 1 & 0 & 0 & 0 & 0 & 1 & 0 \\
0 & 1 & 0 & 0 & 0 & 1 & 0 & 1 & 0 & 0 & 0 \\
1 & 0 & 0 & 0 & 1 & 0 & 0 & 0 & 0 & 1 & 0 \\
1 & 0 & 0 & 0 & 0 & 1 & 0 & 1 & 0 & 0 & 0
\end{pmatrix}
\begin{pmatrix}
x_0 \\
x_1 \\
x_{10} \\
x_2 \\
x_3 \\
x_4 \\
x_5 \\
x_6 \\
x_7 \\
x_8 \\
x_9
\end{pmatrix}
\stackrel{\circ}{=}
\begin{pmatrix}
\nu_\ominus(a) \\
\nu_\oplus(a) \\
\nu_\ominus(a_{rec}) \\
\nu_\oplus(a_{rec}) \\
\nu_\ominus(signvar) \\
\nu_\oplus(signvar) \\
\nu_\ominus(s) \\
\nu_\oplus(s) \\
\nu_\ominus(s_{rec}) \\
\nu_\oplus(s_{rec})
\end{pmatrix}
$$

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
