# Peer review of "Exact Boolean Abstraction of Linear Equation Systems"

_computation, doi:10.3390/computation9110113_

Round 1
Reviewer 1 Report
Comments: This paper studies the problem of how to compute the Boolean abstraction of the solution set of a linear equation system over the positive reals. The authors presented a new rewriting algorithm that makes linear equation systems exact for the boolean abstraction while preserving the solutions over the positive reals. The computation of the elementary modes may require exponential time in the worst case, but is often feasible in practice with freely available tools.
The idea and method are new and interesting, the main results are well presented, the paper is well organized, and also provides a new insight of computing the sign abstraction of linear equation systems. There are some minor concerns that should be carefully addressed by the authors:
\begin{enumerate}
\item[(1)] In this paper, there are many theoretical results in each section, thus it would be nice to rearrange the main results of this paper to highlight the contributions, in order to improve the readability for authors.
\item[(2)] Some mathematical symbols are used without any definitions, please check the whole paper and also the symbols to present a clear presentations.
\item[(3)] In this paper, there are many definitions and proposition, it would be desirable for the authors to make some discussion with some existing method.
\item[(4)] There are many typos and grammatical errors in this paper, which result in the difficulty when reading this manuscript. In addition, the format of the references in this paper is not uniform, please change them carefully, see [10], [16].
\end{enumerate}
Author Response
- In this paper, there are many theoretical results in each section, thus it would be nice to rearrange the main results of this paper to highlight the contributions, in order to improve the readability for authors.
We rearranged the presentation of the contributions in the introduction.
- In order to structure our approach more clearly, we introduced subsection and paragraphs.
- This subsection on the contributions should become clearer with the added structures and subtitles.
- The main Theorem 6 is now pointed out already in the contributions subsection of the introduction.
- The other novel Theorems 2, 3, 4, 6 are cited and explained on high level in the subsection on the contributions of the introduction. - Some mathematical symbols are used without any definitions, please check the whole paper and also the symbols to present a clear presentations.
- We made the introduction lighter, avoiding the use of any notation that wasn’t introduced earlier. Now all notations – standard or not – are introduced before they are used.
- We also completed few missing standard notions in the preliminary section 2. - In this paper, there are many definitions and proposition, it would be desirable for the authors to make some discussion with some existing method.
- We distinguished a subsection on related work in the introduction.
- We discuss own related work more clearly
- We added a paragraph on related work on abstract interpretation with the polyhedron abstract domain. - There are many typos and grammatical errors in this paper, which result in the difficulty when reading this manuscript.
We did our best to fix them. For sure we fixed all typos listed by some of the reviewers. But we fixed also some others, that we spotted during the revision. - In addition, the format of the references in this paper is not uniform, please change them carefully, see [10], [16].
We adapted the bibtex entries of the preprints [16] and [3] which were missing online references by URLs. We also completed some other conference publications such as [2], [6], [7], [12], [16], [17] ... by adding some details such as doi identifiers whenever available. This was done by chosing bibtex entries from DBLP rather than HAL. Otherwise, the bibliography is generated automatically by bibtex, which should
rule out other uniformity problems (except for incompleteness and duplicates).
Reviewer 2 Report
The paper presents a series of theoretical results which together comprise a method for construction of *exact* Boolean abstractions of systems of linear equations over positive reals. This is then extended to a sign-abstraction over reals by decomposition of one (possibly negative) variable into two positive ones.
Originality: The approach is highly reminiscent of the previous work done by the authors in chemical reaction networks (see [1] and [2], however note that [1] is "only" a preprint as far as I know). These papers contain the exact Boolean abstraction that is also used here, however differ in terms of applications: here, the authors show how to use this technique to obtain a sign abstraction of the linear system which can be used, for example, in program analysis.
So overall, there is a novel contribution and the authors declare the existence of previous work, however I do believe that the paper could communicate better which parts are novel and which are adapted from previous papers for completeness/readability.
Presentation: The paper is rather heavy and technical, but I haven't encountered any issues with soundness and could follow the author's logical reasoning without major problems. That being said, especially the Introduction section is in my opinion too technical. Especially because it uses terms and notations which are only defined later in the paper without sufficient high-level explanation (and this problem slightly also extends to the preliminaries, where \Sigma_{arith}-structure appears before it is defined, etc.). I understand that some of this is standard notation which the reader is expected to know, but I would still appreciate a more high-level overview of the paper and significance of the results in this section. Another small issue is a higher than expected number of typos (partial list at the end of the review).
Also, this paper is concerned with sign-abstraction (primarily) for program analysis, but the related work contains mainly the most general literature on abstract interpretation. Is there any other related work that specialises in sign-abstractions that could be added here?
Also a small note about proof of Theorem 2: I understand this is from previous work, but I don't find the proof to be very readable. I'd preferably reference it without proof, or a have a more extended proof in the appendix.
Experiments: Finally, the method is demonstrated on a small example of an integral approximation procedure, where it eliminates spurious abstract transitions that prevent us from verifying the safety of the procedure. The example is nice as a demonstration, but I do wonder how typical are the properties that the method requires (linearity) in actual programs.
[1] Allart, Emilie, Joachim Niehren, and Cristian Versari. "Reaction Networks to Boolean Networks: Exact Boolean Abstraction for Linear Equation Systems." (2021).
[3] Emilie Allart, Joachim Niehren, Cristian Versari, Computing difference abstractions of linear equation systems, Theoretical Computer Science, 2021
- Reference [7] and [17] are the same paper.
- 57: The satisfiability *of* systems
- 66: this *is* not always possible
- 142: Let \Sigma... *be* the arithmetic signature
- 143: all other *operators are* binary function symbols
- 153: So we *must see*
- 228: same arity *as* y
- Definition 12: missing space in h:S
- Lemma 15: missing comma in tuple
- 325: *a* natural number
- 443: natural *numbers* n \in N
- Proposition 28 has a very very long list of *and* clauses that is hard to follow.
- 483: *systems* are
- 512: *without* loss of generality?
- 668: we now *have*
- 671: sign abstraction of a *system*
- 705: set of *solutions*
Author Response
So overall, there is a novel contribution and the authors declare the existence of previous work, however I do believe that the paper could communicate better which parts are novel and which are adapted from previous papers for completeness/readability.
We distinguished the subsection on related work. In particularly it permits us to
compare to our own related work more clearly. This is a little delicate, since some of the results found later were published earlier and so depend on the contribution presented here, while others are yet unpublished.
- the previous work [6] is discussed in the paragraph of related work on “Change
Prediction of Reaction Networks”
- the unpublished preprint [17] is discussed in the paragraph on related work on "
Abstracting Reaction Networks to Boolean Networks “
- when it comes to implementation and experimentation, the novel contributions
and the previous results should be clarified.
The Introduction section is in my opinion too technical. Especially because it uses terms and notations which are only defined later in the paper without sufficient high-level explanation (and this problem slightly also extends to the preliminaries, where \Sigma_{arith}-structure appears before it is defined, etc.). I understand that some of this is standard notation which the reader is expected to know, but I would still appreciate a more high-level overview of the paper and significance of the results in this section.
We revised the introduction entirely in order to make the presentation more high-
level.
- the technical notations that were not explained previously are removed.
- missing standard notation is now always introduced before used also in the preliminaries.
Another small issue is a higher than expected number of typos (partial list at the end of the review).
We did our best to fix them. For sure we fixed all typos listed by some of the
reviewers. But we fixed also some others, that we spotted during the revision.
Also, this paper is concerned with sign-abstraction (primarily) for program analysis, but the related work contains mainly the most general literature on abstract interpretation. Is there any other related work that specialises in sign-abstractions that could be added here?
We added a paragraph on abstract interpretation over relational abstract
domains. We argue that abstract program interpretation over the polyhedral domain is particularly relevant to our results, since it may produce linear equations systems as a result, for which one then might want to compute the sign abstraction.
Also a small note about proof of Theorem 2: I understand this is from previous work, but I don't find the proof to be very readable. I'd preferably reference it without proof, or a have a more extended proof in the appendix.
We revised the proof of the elementary mode theorem, which is now Theorem 3. For this we formulated the previously known aspects into the Folklore Theorem 2 that we added.
The example is nice as a demonstration, but I do wonder how typical are the properties that the method requires (linearity) in actual programs.
We now argue the generality of the approach in a paragraph added to section 10. While not being always exact, it can always be used to reduce the overapproximation.
Reviewer 3 Report
Тhe proposed manuscript is devoted to the problem of possible computation of the boolean abstraction of the solution set of a linear equation system over the positive reals. The authors present a new rewriting algorithm that makes linear equation systems exact for the boolean abstraction while preserving the solutions over the positive reals. The proposed algorithm is based on the elementary modes of the linear equation system.
Preliminary notations and concepts are reviewed in detail. The presentation of the main results is clear and comprehensive. From a formal point of view, all the contents seems to be correct and proofs complete. The results are valuable and worthy of being published taking into account their possible applications for analysis of function programs with linear arithmetics. of chemical reaction networks, for change prediction algorithms for flux networks in systems biology etc.
Minor revisions are suggested to improve the quality of the exposition:
p. 1, line 26: It should be “variable [4,5].” instead of “variable. [4,5].”
p. 2, line 66: I suggest to write “there may be infinitely many such consequences, so that” instead of “there may be infinitely many, so that”
p. 4, line 139: It should be “the set natural numbers” instead of “the set of natural numbers”
Author Response
All the suggested corrections have been applied, where the text has not been rewritten completely during the revision.